# DIFFERENTIAL PRIVATE ONE PERMUTATION HASHING

## ABSTRACT

Minwise hashing (MinHash) is a standard hashing algorithm for large-scale search and learning with the binary Jaccard similarity. One permutation hashing (OPH) is an effective and efficient alternative of MinHash which splits the data into $K$ bins and generates hash values within each bin. In this paper, to protect the privacy of the output sketches, we combine differential privacy (DP) with OPH, and propose DP-OPH framework with three variants: DP-OPH-fix, DP-OPH-re and DP-OPH-rand, depending on the densification strategy to deal with empty bins in OPH. Detailed algorithm design and privacy and utility analysis are provided. The proposed DP-OPH methods significantly improves the DP minwise hashing (DP-MH) alternative in the literature. Experiments on similarity search confirm the effectiveness of our proposed algorithms. We also provide an extension to real-value data, named DP-BCWS, in the appendix.

## 1 INTRODUCTION

Let $\boldsymbol{u}, \boldsymbol{v} \in \{0,1\}^D$ be two $D$-dimensional binary vectors. In this paper, we focus on the hashing algorithms for the Jaccard similarity (a.k.a. the "resemblance") defined as $J(\boldsymbol{u}, \boldsymbol{v}) = \frac{\sum_{i=1}^{D} \mathbb{1}\{\boldsymbol{u}_i=\boldsymbol{v}_i=1\}}{\sum_{i=1}^{D} \mathbb{1}\{\boldsymbol{u}_i+\boldsymbol{v}_i\geq 1\}}$. This is a widely used similarity measure in machine learning applications. $\boldsymbol{u}$ and $\boldsymbol{v}$ can also be viewed as two sets of items represented by the locations of non-zero entries. In industrial applications with massive data size, directly calculating the pairwise Jaccard similarity among the data points becomes too expensive. To accelerate large-scale search and learning, the celebrated *"minwise hashing"* (MinHash) algorithm (Broder, 1997; Broder et al., 1997) has been a standard hashing technique for approximating the Jaccard similarity in massive binary datasets. It has seen numerous applications such as near neighbor search, duplicate detection, malware detection, clustering, large-scale learning, social networks, and computer vision (Indyk & Motwani, 1998; Charikar, 2002; Fetterly et al., 2003; Das et al., 2007; Buehrer & Chellapilla, 2008; Bendersky & Croft, 2009; Chierichetti et al., 2009; Pandey et al., 2009; Lee et al., 2010; Deng et al., 2012; Chum & Matas, 2012; Tamersoy et al., 2014; Shrivastava & Li, 2014; Zhu et al., 2017; Nargesian et al., 2018; Wang et al., 2019; Lemiesz, 2021; Feng & Deng, 2021; Li & Li, 2022). The output of MinHash is an integer. For large-scale applications, to store and use the hash values (or called sketches) more conveniently and efficiently, Li & König (2010) proposed $b$-bit MinHash that only stores the last $b$ bits of the hashed integers, which is memory-efficient and convenient for similarity search and machine learning. Thus, it has been a popular coding strategy for the MinHash values and its alternatives (Li et al., 2011; 2015; Shah & Meinshausen, 2017; Yu & Weber, 2022).

### 1.1 ONE PERMUTATION HASHING (OPH) FOR JACCARD SIMILARITY APPROXIMATION

To use MinHash in practice, we need to generate $K$ hash values to achieve good utility. This requires applying $K$ random permutations (or hash functions as approximations) per data point, yielding an $O(Kf)$ complexity where $f$ is the number of non-zero elements. One permutation hashing (OPH) (Li et al., 2012) provides a strategy to significantly reduce the complexity to $O(f)$. The idea of OPH is: to generate $K$ hashes, we split the data vector into $K$ non-overlapping bins, and conduct MinHash within each bin. Yet, empty bins may arise which breaks the alignment of the hashes. To deal with empty bins, densification schemes (Shrivastava, 2017; Li et al., 2019) are proposed that fill the empty bins with some non-empty bin. It is shown that OPH with densification provides unbiased Jaccard estimator, and the estimation variance can often be smaller than that of MinHash. OPH has been widely used as an improved method over MinHash for the Jaccard similarity (Dahlgaard et al., 2017; Zhao et al., 2020; Jia et al., 2021; Tseng et al., 2021; Jiang et al., 2022).

## 1.2 HASHING/SKETCHING AND DIFFERENTIAL PRIVACY

MinHash and OPH both belong to the broad family of probabilistic hashing/sketching methods designed for various purposes and tasks. Examples of more sketching methods include the random projection (RP) based methods for cosine estimation (Charikar, 2002; Vempala, 2005), the count-sketch (CS) for frequency estimation (Charikar et al., 2004), and the Flajolet-Martin (FM) sketch (Flajolet & Martin, 1985) and HyperLogLog sketch (Flajolet et al., 2007) for cardinality estimation, etc. Since the data sketches produce "summaries" of the original data, sketching/hashing may also cause data privacy leakage. Therefore, protecting the privacy of the data sketches has become an important topic which has gained growing research interests in recent years.

Differential privacy (DP) (Dwork et al., 2006b) has become a popular privacy definition with rigorous mathematical formulation, which has been widely applied to clustering, regression and classification, principle component analysis, matrix completion, optimization, deep learning (Blum et al., 2005; Chaudhuri & Monteleoni, 2008; Feldman et al., 2009; Gupta et al., 2010; Chaudhuri et al., 2011; Kasiviswanathan et al., 2013; Zhang et al., 2012; Abadi et al., 2016; Agarwal et al., 2018; Ge et al., 2018; Wei et al., 2020; Dong et al., 2022), etc. Prior efforts have also been conducted on combining differential privacy with the aforementioned hashing algorithms, e.g., for RP (Blocki et al., 2012; Kenthapadi et al., 2013; Stausholm, 2021), count-sketch (Zhao et al., 2022), and FM sketch (Smith et al., 2020; Dickens et al., 2022). Some works (e.g., Blocki et al. (2012); Smith et al. (2020); Dickens et al. (2022)) assumed "internal randomness", i.e., the randomness of the hash functions are kept private, and showed that many hashing methods themselves already possess strong DP property under some data conditions. However, this setting is more restrictive in practice as it requires that the hash keys or projection matrices cannot be accessed by any adversary. In another setup (e.g., Kenthapadi et al. (2013); Stausholm (2021); Zhao et al. (2022)), both the randomness of the hash functions and the algorithm outputs are treated as public information, and perturbation mechanisms are developed to make the algorithms differentially private.

**Contributions.** While prior works have proposed DP algorithms for some sketching methods mentioned earlier, the differential privacy of OPH and MinHash for the Jaccard similarity has not been well studied. In this paper, we mainly focus on the differential privacy of one permutation hashing (OPH), the state-of-the-art framework for hashing the Jaccard similarity. We consider the more practical and general setup where the randomness of the algorithm is external/public.

We develop three variants under the DP-OPH framework, DP-OPH-fix, DP-OPH-re, and DP-OPH-rand, corresponding to fixed densification, re-randomized densification, and no densification for OPH, respectively. We provide detailed algorithm design and privacy analysis for each variant, and compare them with a DP MinHash (DP-MH) method. In our retrieval experiments, we show that the proposed DP-OPH method substantially improves DP-MH, and re-randomized densification is superior over fixed densification in terms of differential privacy. DP-OPH-rand performs the best when $\epsilon$ is small, while DP-OPH-re is the most performant in when larger $\epsilon$ is allowed. We also extend our algorithms to real-value datasets and develop DP-BCWS algorithm in Appendix A.

## 2 BACKGROUND: MINHASH, $b$-BIT CODING, AND DIFFERENTIAL PRIVACY

---

**Algorithm 1** Minwise hashing (MinHash)

---

**Input:** Binary vector $\boldsymbol{u} \in \{0, 1\}^D$; number of hash values $K$
**Output:** $K$ MinHash values $h_1(\boldsymbol{u}), ..., h_K(\boldsymbol{u})$
  1: Generate $K$ independent permutations $\pi_1, ..., \pi_K$: $[D] \to [D]$ with seeds $1, ..., K$ respectively
  2: **for** $k = 1$ to $K$ **do**
  3:     $h_k(\boldsymbol{u}) \leftarrow \min_{i:u_i \neq 0} \pi_k(i)$
  4: **end for**

---

**Minwise hashing (MinHash).** The MinHash method is summarized in Algorithm 1. We first generate $K$ independent permutations $\pi_1, ..., \pi_K : [D] \mapsto [D]$. Here, $[D]$ denotes $\{1, ..., D\}$. For each permutation, the hash value is the first non-zero location in the permuted vector, i.e., $h_k(\boldsymbol{u}) = \min_{i:v_i \neq 0} \pi_k(i)$, $\forall k = 1, ..., K$. Analogously, for another data vector $\boldsymbol{v} \in \{0, 1\}^D$, we also obtain

$K$ hash values, $h_k(\boldsymbol{v})$. The MinHash estimator of $J(\boldsymbol{u}, \boldsymbol{v})$ is the average over the hash collisions:

$$\hat{J}_{MH}(\boldsymbol{u}, \boldsymbol{v}) = \frac{1}{K} \sum_{k=1}^{K} \mathbb{1}\{h_k(\boldsymbol{u}) = h_k(\boldsymbol{v})\}, \tag{1}$$

where $\mathbb{1}\{\cdot\}$ is the indicator function. By standard probability calculation, we can show that $\mathbb{E}[\hat{J}_{MH}] = J$ and $Var[\hat{J}_{MH}] = \frac{J(1-J)}{K}$. In practice, $K$ does not need to be very large to achieve good utility. For instance, usually $128 \sim 1024$ hash values would be sufficient for search and learning problems (Indyk & Motwani, 1998; Li et al., 2011; Shrivastava & Li, 2014).

$b$-**bit coding of the hash value.** Li & König (2010) proposed "$b$-bit minwise hashing" as a convenient coding strategy for the integer hash value $h(\boldsymbol{u})$ generated by MinHash (or by OPH which will be introduced later). Basically, we only keep the last $b$-bits of each hash value. In our analysis, for convenience, we assume that "taking the last $b$-bits" can be achieved by some "rehashing" trick to map the integer values onto $\{0, ..., 2^b - 1\}$ uniformly. There are at least three benefits of this coding strategy: (i) storing only $b$ bits saves the storage cost compared with storing the full 32 or 64 bit integers; (ii) the last few bits are more convenient for the purpose of indexing, e.g., in approximate nearest neighbor search (Indyk & Motwani, 1998); (iii) we can transform the last few bits into a positional representation, allowing us to approximate the Jaccard similarity by inner product, which is required by training large-scale linear models (Li et al., 2011). Given these advantages, in this work, we will adopt this $b$-bit coding strategy in our private algorithm design.

**Differential privacy (DP).** We formally define differential privacy (DP) as follows.

**Definition 2.1** (Differential privacy (Dwork et al., 2006b)). *For a randomized algorithm* $\mathcal{M} : \mathcal{U} \mapsto Range(\mathcal{M})$ *and* $\epsilon, \delta \geq 0$*, if for any two neighboring datasets* $U$ *and* $U'$*, the following holds*

$$Pr[\mathcal{M}(U) \in Z] \leq e^{\epsilon} Pr[\mathcal{M}(U') \in Z] + \delta$$

*for* $\forall Z \subset Range(\mathcal{M})$*, then algorithm* $\mathcal{M}$ *is said to satisfy* $(\epsilon, \delta)$*-DP. If* $\delta = 0$*,* $\mathcal{M}$ *is called* $\epsilon$*-DP.*

Intuitively, DP requires that the distributions of the outputs before and after a small change in the data are close, so that an adversary cannot detect the change based on the outputs. Smaller $\epsilon$ and $\delta$ implies stronger privacy. The parameter $\delta$ is usually interpreted as the "failure probability" allowed for the $\epsilon$-DP guarantee to be violated.

**Privacy statement and applications.** We follow the standard attribute-level DP setup in aforementioned related works on DP hashing/sketching: $\boldsymbol{u}, \boldsymbol{u}' \in \{0, 1\}^D$ are called neighboring if they differ in one dimension. Treating the binary vectors as sets, with our proposed DP-OPH algorithms, *an adversary cannot detect from the output sketches whether any item exists in the set or not, which holds independently for all the data vectors in the database*. DP-OPH can naturally be applied as a private variant of OPH in cases where MinHash-type methods are found to be useful. As a concrete example application, the bioinformatics community releases sets of MinHashes for all known genomes on a regular basis (Ondov et al., 2016; Brown & Irber, 2016), which are used for downstream ML tasks like similarity search, classification, clustering, etc. (Berlin et al., 2015) In this type of data, each data point corresponds to (a large set of) genes of a human, which contains the biological information of an individual which is highly sensitive and confidential. Our methods protect the identification of any gene from the released sketches in the DP sense.

## 3 HASHING FOR JACCARD SIMILARITY WITH DIFFERENTIAL PRIVACY

In this section, we present DP-OPH algorithms based on privatizing the $b$-bit hash values from OPH and utility analysis, and demonstrate its advantage over a DP-MinHash alternative.

### 3.1 ONE PERMUTATION HASHING (OPH)

Algorithm 2 outlines the steps of OPH: we first use a permutation $\pi$ (same for all data vectors) to randomly split the feature dimensions $[D]$ into $K$ bins $\mathcal{B}_1, ..., \mathcal{B}_K$ with equal length $d = D/K$ (assuming integer division holds). Then, for each bin $\mathcal{B}_k$, we set the smallest permuted index of "1" as the $k$-th OPH hash value. If $\mathcal{B}_k$ is empty (i.e., it does not contain any "1"), we record an "$E$" representing empty bin. Li et al. (2012) showed that we can construct statistically unbiased Jaccard estimators by ignoring the empty bins. However, this estimator is unstable when the data is relatively

---

**Algorithm 2** One Permutation Hashing (OPH)

---

**Input:** Binary vector $\boldsymbol{u} \in \{0, 1\}^D$; number of hash values $K$
**Output:** $K$ OPH hash values $h_1(\boldsymbol{u}), ..., h_K(\boldsymbol{u})$

1: Let $d = D/K$. Use a permutation $\pi : [D] \mapsto [D]$ with fixed seed to randomly split $[D]$ into $K$ equal-size bins $\mathcal{B}_1, ..., \mathcal{B}_K$, with $\mathcal{B}_k = \{j \in [D] : (k-1)d + 1 \leq \pi(j) \leq kd\}$
2: **for** $k = 1$ to $K$ **do**
3:     **if** Bin $\mathcal{B}_k$ is non-empty **then**
4:         $h_k(\boldsymbol{u}) \leftarrow \min_{j \in \mathcal{B}_k, u_j \neq 0} \pi(j)$
5:     **else**
6:         $h_k(\boldsymbol{u}) \leftarrow E$
7:     **end if**
8: **end for**

---

**Algorithm 3** OPH-fix and OPH-re: OPH with fixed and re-randomized densification

---

**Input:** OPH hash values $h_1(\boldsymbol{u}), ..., h_K(\boldsymbol{u})$ each in $[D] \cup \{E\}$; bins $\mathcal{B}_1, ..., \mathcal{B}_K$; $d = D/K$
**Output:** $K$ densified OPH hash values $h_1(\boldsymbol{u}), ..., h_K(\boldsymbol{u})$

1: Let $NonEmptyBin = \{k \in [K] : h_k(\boldsymbol{u}) \neq E\}$
2: **for** $k = 1$ to $K$ **do**
3:     **if** $h_k(\boldsymbol{u}) = E$ **then**
4:         Uniformly randomly select $k' \in NonEmptyBin$
5:         $h_k(\boldsymbol{u}) \leftarrow h_{k'}(\boldsymbol{u})$                ▷ fixed densification
6:         **Or**
7:         $MapToIndex = SortedIndex\left(\pi(\mathcal{B}_k)\right) + (k'-1)d$
8:         $\pi^{(k)} : \pi(\mathcal{B}_{k'}) \mapsto MapToIndex$      ▷ within-bin partial permutation
9:         $h_k(\boldsymbol{u}) \leftarrow \min_{j \in \mathcal{B}_{k'}, u_j \neq 0} \pi^{(k)}\left(\pi(j)\right)$    ▷ re-randomized densification
10:     **end if**
11: **end for**

---

sparse; moreover, since empty bins are different for every distinct data vector, the vanilla OPH hash values do not form a metric space (i.e., the hash values of different data points are not aligned).

**Densification for OPH.** To tackle the issue caused by empty bins, a series of works has been conducted to densify the OPH. The general idea is to "borrow" the data/hash from non-empty bins, with some careful design. In Algorithm 3, we present two recent representatives of OPH densification methods: fixed densification (Shrivastava, 2017) and re-randomized densification (Li et al., 2019), noted as OPH-fix and OPH-re, respectively. Given an OPH hash vector from Algorithm 2 (possibly containing "$E$"s), we denote the set of non-empty bins $NonEmptyBin = \{k \in [K] : h_k(\boldsymbol{u}) \neq E\}$. The densification procedure scans over $k = 1, ..., K$. For each $k$ with $h_k(\boldsymbol{u}) = E$, we do:

1. Uniformly randomly pick a bin $k' \in NonEmptyBin$ that is non-empty.

2. (a) OPH-fix: we directly copy the $k'$-th hash value: $h_k(\boldsymbol{u}) \leftarrow h_{k'}(\boldsymbol{u})$.
   (b) OPH-re: we apply an additional minwise hashing to bin $\mathcal{B}_{k'}$ using the "partial permutation" of $\mathcal{B}_k$ to get the hash for $h_k(\boldsymbol{u})$.

Specifically, In Algorithm 2, for re-randomized densification, $SortedIndex$ and $MapToIndex$ are used to define the within bin "partial permutation" $\pi^{(k)}$ of bin $\mathcal{B}_k$ for re-randomizing the empty bins.

It is shown that for both variants, the Jaccard estimator of the same form as (1) is unbiased. Li et al. (2019) showed that re-randomized densification always achieves smaller Jaccard estimation variance than that of fixed densification, and the improvement is especially significant when the data is sparse. Similar to $b$-bit MinHash, we can also keep the last $b$ bits of the OPH hash values for convenient use.

### 3.2 Differential Private One Permutation Hashing (DP-OPH)

**DP-OPH with densification.** To privatize densified OPH, in Algorithm 4, we first take the last $b$ bits of the hash values. Since the output space is finite with cardinality $2^b$, we apply the randomized response technique (Dwork & Roth, 2014; Wang et al., 2017) to flip the bits to achieve DP. After

---

**Algorithm 4** Differentially Private Densified One Permutation Hashing (DP-OPH-fix, DP-OPH-re)

---

**Input:** Densified OPH hash values $h_1(\boldsymbol{u}), ..., h_K(\boldsymbol{u})$; number of bits $b$; $\epsilon > 0, 0 < \delta < 1$
$\qquad f$: lower bound on the number of non-zeros in each data vector
**Output:** $b$-bit DP-OPH values $\tilde{h}(\boldsymbol{u}) = [\tilde{h}_1(\boldsymbol{u}), ..., \tilde{h}_K(\boldsymbol{u})]$
1: Take the last $b$ bits of all hash values $\qquad\qquad\qquad\qquad\triangleright$ After which $h_k(\boldsymbol{u}) \in \{0, ..., 2^b - 1\}$
2: Set $N = F_{fix}^{-1}(1-\delta; D, K, f)$ (for DP-OPH-fix) or $N = F_{re}^{-1}(1-\delta; D, K, f)$ (for DP-OPH-re), and let $\epsilon' = \epsilon/N$
3: **for** $k = 1$ to $K$ **do**
4: $\qquad \tilde{h}_k(\boldsymbol{u}) = \begin{cases} h_k(\boldsymbol{u}), & \text{with prob. } \frac{e^{\epsilon'}}{e^{\epsilon'}+2^b-1} \\ i, & \text{with prob. } \frac{1}{e^{\epsilon'}+2^b-1}, \text{for } i \in \{0, ..., 2^b - 1\}, \ i \neq h_k(\boldsymbol{u}) \end{cases}$
5: **end for**

---

running Algorithm 3, suppose a densified OPH hash value $h_k(\boldsymbol{u}) = j$, $j \in 0, ..., 2^b - 1$. With some $\epsilon' > 0$ that will be specified later, we output $\tilde{h}_k(\boldsymbol{u}) = j$ with probability $\frac{e^{\epsilon'}}{e^{\epsilon'}+2^b-1}$, and $\tilde{h}_k(\boldsymbol{u}) = i$ for $i \neq j$ with probability $\frac{1}{e^{\epsilon'}+2^b-1}$. It is easy to verify that, for a neighboring data $\boldsymbol{u}'$, when $h_k(\boldsymbol{u}') = j$, for $\forall i \in 0, ..., 2^b - 1$, we have $\frac{P(\tilde{h}_k(\boldsymbol{u})=i)}{P(\tilde{h}_k(\boldsymbol{u}')=i)} = 1$; when $h_k(\boldsymbol{u}') \neq j$, we have $e^{-\epsilon'} \leq \frac{P(\tilde{h}_k(\boldsymbol{u})=i)}{P(\tilde{h}_k(\boldsymbol{u}')=i)} \leq e^{\epsilon'}$. Therefore, for a single hash value, this bit flipping satisfies $\epsilon'$-DP.

It remains to determine $\epsilon'$. Naively, since the perturbations (flipping) of the hash values are independent, by the composition property of DP (Dwork et al., 2006a), simply setting $\epsilon' = \epsilon/K$ for all $K$ MinHash values would achieve overall $\epsilon$-DP (for the hashed vector). However, since $K$ is usually around hundreds, a very large $\epsilon$ value is required for this strategy to be useful. To this end, we can trade a small $\delta$ in the DP definition for a significantly reduced $\epsilon$. Note that, not all the $K$ hashed bits will change after we switch from $\boldsymbol{u}$ to its neighbor $\boldsymbol{u}'$. Assume each data vector contains at least $f$ non-zeros, which is realistic since many data in practice have both high dimensionality $D$ as well as many non-zero elements. Intuitively, when the data is not too sparse, $\boldsymbol{u}$ and $\boldsymbol{u}'$ tends to be similar. Therefore, the number of different hash values from Algorithm 3, $X = \sum_{k=1}^K \mathbb{1}\{h_k(\boldsymbol{u}) \neq h_k(\boldsymbol{u}')\}$, can be upper bounded by some $N$ with probability $1 - \delta$. In the proof, this allows us to set $\epsilon' = \epsilon/N$ in the flipping probability and count $\delta$ as the failure probability in $(\epsilon, \delta)$-DP.

Next, we derive the distribution of $X$. Accordingly, in Algorithm 4, we set $N = F_{fix}^{-1}(1-\delta; D, f, K)$ for DP-OPH-fix, $N = F_{re}^{-1}(1 - \delta; D, f, K)$ for DP-OPH-re, where $F_{fix}(x) = P(X \leq x)$ is the cumulative mass function (CMF) of $X$ with OPH-fix ((2) + (3)), and $F_{re}$ is the CMF of $X$ with OPH-re ((2) + (4)), and $F^{-1}$ is the inverse CMF. The proof can be found in Appendix B.

**Lemma 3.1.** *Let $\boldsymbol{u}, \boldsymbol{u}' \in \{0, 1\}^D$ be neighbors. Denote $X = \sum_{k=1}^K \mathbb{1}\{h_k(\boldsymbol{u}) \neq h_k(\boldsymbol{u}')\}$ where the hashes are generated by Algorithm 3. Denote $f = |\boldsymbol{u}|, d = D/K$. We have*

$$P(X = x) = \sum_{j=\max(0, K-f)}^{K-\lceil f/d \rceil} \sum_{z=1}^{\min(f,d)} \Theta(x, j, z | K), \quad for \ x = 0, ..., K - \lceil f/d \rceil, \qquad (2)$$

*with $\Theta(x, j, z | K) = \tilde{P}(x|z, j) P\left(\tilde{f} = z | K - j\right) P(N_{emp} = j)$, where $P\left(\tilde{f} = z | K - j\right)$ is given in Lemma B.2, and $P(N_{emp} = j)$ is from Lemma B.1. Moreover,*

$$\textbf{For OPH-fix:} \ \ \tilde{P}(x|z, j) = \mathbb{1}\{x = 0\}\left(1 - P_{\neq}\right) + \mathbb{1}\{x > 0\}P_{\neq} \cdot g_{bino}\left(x - 1; \frac{1}{K-j}, j\right), \qquad (3)$$

$$\textbf{For OPH-re:} \ \ \tilde{P}(x|z, j) = \left(1 - P_{\neq}\right) \cdot g_{bino}\left(x; \frac{P_{\neq}}{K-j}, j\right) + P_{\neq} \cdot g_{bino}\left(x - 1; \frac{P_{\neq}}{K-j}, j\right), \qquad (4)$$

*where $g_{bino}(x; p, n)$ is the CMF of $Binomial(p, n)$, and $P_{\neq}(z, b) = \left(1 - \frac{1}{2^b}\right)\frac{1}{z}$.*

The privacy guarantee of DP-OPH with densification is shown as below.

**Theorem 3.2.** *Both DP-OPH-fix and DP-OPH-re in Algorithm 4 achieve $(\epsilon, \delta)$-DP.*

---

**Algorithm 5** Differentially Private One Permutation Hashing with Random Bits (DP-OPH-rand)

---

**Input:** OPH hash values $h_1(\boldsymbol{u}), ..., h_K(\boldsymbol{u})$ from Algorithm 2; number of bits $b$; $\epsilon > 0$
**Output:** DP-OPH-rand hash values $\tilde{h}(\boldsymbol{u}) = [\tilde{h}_1(\boldsymbol{u}), ..., \tilde{h}_K(\boldsymbol{u})]$

1: Take the last $b$ bits of all hash values $\qquad\qquad\qquad$ ▷ After which $h_k(\boldsymbol{u}) \in \{0, ..., 2^b - 1\}$
2: **for** $k = 1$ to $K$ **do**
3: $\quad$ **if** $h_k(\boldsymbol{u}) \neq E$ **then**
4: $\qquad \tilde{h}_k(\boldsymbol{u}) = \begin{cases} h_k(\boldsymbol{u}), & \text{with prob. } \frac{e^\epsilon}{e^\epsilon + 2^b - 1} \\ i, & \text{with prob. } \frac{1}{e^{\epsilon'} + 2^b - 1}, \text{ for } i \in \{0, ..., 2^b - 1\}, \ i \neq h_k(\boldsymbol{u}) \end{cases}$
5: $\quad$ **else**
6: $\qquad \tilde{h}_k(\boldsymbol{u}) = i$ with probability $\frac{1}{2^b}$, for $i = 0, ..., 2^b - 1$ ▷ Assign random bits to empty bin
7: $\quad$ **end if**
8: **end for**

---

**Algorithm 6** Differentially Private MinHash (DP-MH)

---

**Input:** MinHash values $h_1(\boldsymbol{u}), ..., h_K(\boldsymbol{u})$; number of bits $b$; $\epsilon > 0, 0 < \delta < 1$
$\qquad$ $f$: lower bound on the number of non-zeros in each data vector
**Output:** DP-MH values $\tilde{h}(\boldsymbol{u}) = [\tilde{h}_1(\boldsymbol{u}), ..., \tilde{h}_K(\boldsymbol{u})]$

1: Take the last $b$ bits of all hash values $\qquad\qquad\qquad$ ▷ After which $h_k(\boldsymbol{u}) \in \{0, ..., 2^b - 1\}$
2: Set $N = F_{bino}^{-1}(1 - \delta; \frac{1}{f}, K)$, and $\epsilon' = \epsilon/N$
3: **for** $k = 1$ to $K$ **do**
4: $\quad \tilde{h}_k(\boldsymbol{u}) = \begin{cases} h_k(\boldsymbol{u}), & \text{with prob. } \frac{e^{\epsilon'}}{e^{\epsilon'} + 2^b - 1} \\ i, & \text{with prob. } \frac{1}{e^{\epsilon'} + 2^b - 1}, \text{ for } i \in \{0, ..., 2^b - 1\}, \ i \neq h_k(\boldsymbol{u}) \end{cases}$
5: **end for**

---

**DP-OPH without densification.** From the practical perspective, we may also choose to privatize the OPH without densification (i.e., add DP to the output of Algorithm 2). The first step is to take the last $b$ bits of every non-empty hash and get $K$ hash values from $\{0, ..., 2^b - 1\} \cup \{E\}$. Then, for non-empty bins, we keep the hash value with probability $\frac{e^\epsilon}{e^\epsilon + 2^b - 1}$, and randomly flip it otherwise. For empty bins (i.e., $h_k(\boldsymbol{u}) = E$), we simply assign a random value in $\{0, ..., 2^b - 1\}$ to $\tilde{h}_k(\boldsymbol{u})$. The formal procedure of this so-called DP-OPH-rand method is summarized in Algorithm 5.

**Theorem 3.3.** *Algorithm 5 achieves $\epsilon$-DP.*

Compared with Algorithm 4, DP-OPH-rand achieves strict DP with smaller flipping probability (effectively, $N \equiv 1$ in Algorithm 4). This demonstrates the essential benefit of the binning operation in OPH, since the change in one data coordinate will only affect one hash value (if densification is not applied). As a result, the non-empty hash values are less perturbed in DP-OPH-rand than in DP-OPH-fix or DP-OPH-re. But this comes with an extra cost as we have to assign random bits to empty bins which do not provide any useful information, and this extra cost does not diminish as $\epsilon$ increases because the number of empty bins only depends on the data itself and $K$.

**DP-MinHash.** While we have presented our main DP-OPH algorithms, we also present a DP Min-Hash (DP-MH) method (Algorithm 6) as a baseline comparison. The mechanism of DP-MH is similar to that of densified DP-OPH. The difference between Algorithm 6 and Algorithm 4 is in the calculation of $N$. In Algorithm 6, we set $N = F_{bino}^{-1}(1 - \delta; \frac{1}{f}, K)$ where $F_{bino}^{-1}(x; p, n)$ is the inverse cumulative mass function of $Binomial(p, n)$ with $n$ trials and success probability $p$.

**Theorem 3.4.** *Algorithm 6 is $(\epsilon, \delta)$-DP.*

The proof strategy is similar to Theorem 3.2 by noting that $X = \sum_{k=1}^K \mathbb{1}\{h_k(\boldsymbol{u}) \neq h_k(\boldsymbol{u}')\}$ for neighboring $\boldsymbol{u}$ and $\boldsymbol{u}'$ follows $Binomial(\frac{1}{f}, K)$. In a related work, Aumüller et al. (2020) also proposed to apply randomized response to MinHash. However, the authors incorrectly used a tail bound for the binomial distribution (see their Lemma 1) which is only valid for small deviation. In DP, $\delta$ is often very small (e.g., $10^{-6}$), so the large deviation tail bound should be used which is

looser than the one used therein[1]. That said, in their paper, the perturbation is underestimated and their method does not satisfy DP. In our Algorithm 6, we fix it by using the exact probability mass function to compute the tail probability, avoiding any loss due to the concentration bounds.

### 3.3 COMPARISON OF DP-OPH AND DP-MH

We first analyze the mean of the Jaccard estimators and derive unbiased estimators of $J$. To simplify the formula, we assume that $\boldsymbol{u}$ and $\boldsymbol{v}$ have the same "privacy discount factor" $N$ (which implies that $\boldsymbol{u}$ and $\boldsymbol{v}$ have similar sparsity). The results can be easily extended to the general case.

**Theorem 3.5.** *For $u, v \in \{0,1\}^D$, denote $f_u = |\boldsymbol{u}|$, $f_v = |\boldsymbol{v}|$, $a = |\boldsymbol{u} \cap \boldsymbol{v}|$. Suppose $\boldsymbol{u}$ and $\boldsymbol{v}$ have the same privacy discount factor $N$ in Algorithm 4 or Algorithm 6. Then, $J = \frac{a}{f_u + f_v - a}$. Denote $p = \frac{\exp(\epsilon/N)}{\exp(\epsilon/N) + 2^b - 1}$. For DP-OPH-fix, DP-OPH-re, and DP-MH, define $\hat{J} = \frac{1}{K} \sum_{k=1}^{K} \mathbb{1}\{\tilde{h}_k(\boldsymbol{u}) = \tilde{h}_k(\boldsymbol{v})\}$. We have $\mathbb{E}[\hat{J}] = \frac{(2^b p + 1)^2}{2^b (2^b - 1)} J + \frac{1}{2^b}$. Thus, an unbiased estimator is $\hat{J}_{unbias} = \frac{(2^b - 1)(2^b \hat{J} - 1)}{(2^b p - 1)^2}$.*

The variances of the unbiased estimators defined in Theorem 3.5 are given as below.

**Theorem 3.6.** *Define $J_B = J + (1 - J)\frac{1}{2^b}$, $\tilde{J} = \frac{a-1}{f_u + f_v - a - 1}$. Denote $c_1 = p^2 + \frac{(1-p)^2}{2^b - 1}$, and $c_2 = \frac{2p(1-p)}{2^b - 1} + \frac{2^b - 2}{(2^b - 1)^2}(1 - p)^2$. Define $\zeta(m) = \mathbb{E}[\frac{1}{\tilde{f}}|m]$ where the conditional distribution of $\tilde{f}$ is given in Lemma B.2, and:*

$$\tau_{11} = J\tilde{J}, \quad \tau_{10} = J - J\tilde{J}, \quad \tau_{00} = 1 - 2J + J\tilde{J},$$

$$\tau_{11,f}(m) = \frac{1}{m}J + \frac{m-1}{m}J\tilde{J}, \ \tau_{10,f}(m) = \frac{m-1}{m}(J - J\tilde{J}), \ \tau_{00,f}(m) = 1 - (2 - \frac{1}{m})J + \frac{m-1}{m}J\tilde{J},$$

$$\tau_{11,r}(m) = \frac{\zeta(m)}{m}J + \frac{m - \zeta(m)}{m}J\tilde{J}, \ \tau_{10,r}(m) = \frac{m - \zeta(m)}{m}(J - J\tilde{J}),$$

$$\tau_{00,r}(m) = 1 - (2 - \frac{\zeta(m)}{m})J + \frac{m - \zeta(m)}{m}J\tilde{J}.$$

*Further denote $P_{11} = \tau_{11} + \frac{1}{2^{b-1}}\tau_{10} + \frac{1}{2^{2b}}\tau_{00}$, $P_{10} = J_B - P_{11}$, $P_{00} = 1 - 2J_B + P_{11}$, and $(P_{11,f}, P_{10,f}, P_{00,f})$ and $(P_{11,r}, P_{10,r}, P_{00,r})$ analogously by replacing $(\tau_{11}, \tau_{10}, \tau_{00})$ with $(\tau_{11,f}, \tau_{10,f}, \tau_{00,f})$ and $(\tau_{11,r}, \tau_{10,r}, \tau_{00,r})$, respectively. We have for DP-MH:*

$$Var[\hat{J}_{unbias,MH}] = \frac{1}{K}\left(\frac{(2^b p + 1)^2}{(2^b p - 1)^2}J + \frac{2^b - 1}{(2^b p - 1)^2}\right)\left(\frac{(2^b - 1)^2}{(2^b p - 1)^2} - \frac{(2^b p + 1)^2}{(2^b p - 1)^2}J\right).$$

*For DP-OPH: Let $m = K - N_{emp}$ where $N_{emp}$ is distributed as Lemma B.1.*

$$Var[\hat{J}_{unbias,OPH}] = \frac{2^{2b}(2^b - 1)^2}{(2^b p - 1)^4}\left[\frac{1}{K}\left(\frac{(2^b p + 1)^2}{2^b(2^b - 1)}J + \frac{1}{2^b}\right) + \frac{1}{K^2}A - \left(\frac{(2^b p + 1)^2}{2^b(2^b - 1)}J + \frac{1}{2^b}\right)^2\right],$$

$$A = \mathbb{E}_m\left[m(m-1)H_N + (K-m)(K+m-1)H_E\right],$$

*with $H_N = c_1^2 P_{11} + 2c_1 c_2 P_{10} + c_2^2 P_{00}$. For DP-OPH-fix, $H_E = c_1^2 P_{11,f} + 2c_1 c_2 P_{10,f} + c_2^2 P_{00,f}$; for DP-OPH-re, $H_E = c_1^2 P_{11,r} + 2c_1 c_2 P_{10,r} + c_2^2 P_{00,r}$.*

**Comparison: Densified DP-OPH vs. DP-MH.** We show that OPH is a better method than MinHash from the privacy perspective. Firstly, we compare $N$, the privacy discount factor, in DP-OPH-fix, DP-OPH-re, and DP-MH. Smaller $N$ leads to smaller bit flipping probability which benefits the utility. In Figure 1, we plot $N$ vs. $f$, for $D = 1024$, $K = 64$, and $\delta = 10^{-6}$. Similar comparison also holds for other $D, K$ combinations. We observe that $N$ in DP-OPH is typically smaller than that in DP-MH. Moreover, $N$ for DP-OPH-re is consistently smaller than that of DP-OPH-fix. This illustrates that re-randomization in densification is an important step to achieve stronger privacy.

In Figure 2, we plot the empirical MSE of the unbiased estimators. The data is simulated with $f_u = f_v = f$, and $a = f/2$ (see notations in Theorem 3.5). The empirical MSE matches the variances in Theorem 3.6. DP-OPH-re has smaller variance than DP-OPH-fix and DP-MH.

---

[1] For $X$ following a Binomial distribution with mean $\mu$, Aumüller et al. (2020) used the concentration inequality $P(X \geq (1 + \xi)\mu) \leq \exp(-\frac{\xi^2\mu}{3})$, which only holds when $0 \leq \xi \leq 1$. For large deviations (large $\xi$), the valid Binomial tail bound should be $P(X \geq (1 + \xi)\mu) \leq \exp(-\frac{\xi^2\mu}{\xi+2})$.

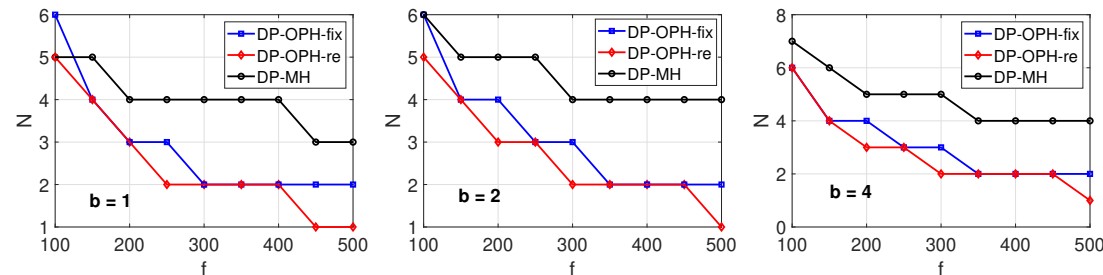

Figure 1: Comparison of the privacy discount factor $N$ for densified DP-OPH and DP-MH, against the number of non-zero elements in the data vector $f$. $D = 1024, K = 64, \delta = 10^{-6}$.

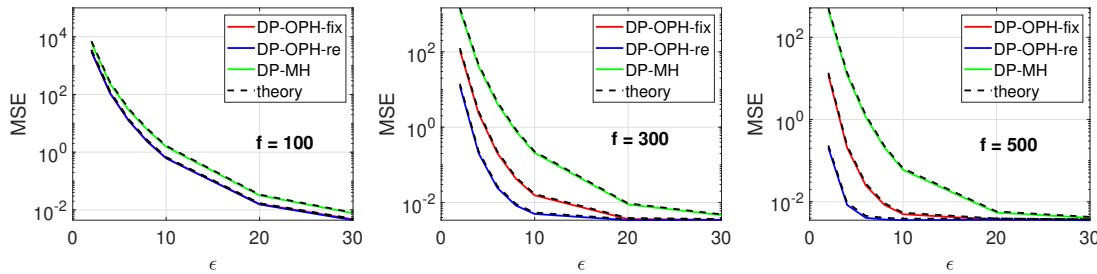

Figure 2: MSE comparison of the unbiased Jaccard estimators (Theorem 3.5). The dash curves are theoretical variances in Theorem 3.6. $D = 1024, K = 64, \delta = 10^{-6}$. $b = 4$.

## 4 EXPERIMENTS

We conduct similarity search on two datasets from genome and web where MinHash-type algorithms are widely used: (1) the Leukemia gene expression dataset (https://sbcb.inf.ufrgs.br/cumida); (2) the Webspam (Chang & Lin, 2011) dataset for spam detection. Both datasets are binarized to 0/1. For Leukemia, we first standardize the features columns (to mean 0 and std 1), and then keep entries larger than 1 to be 1 and zero out the others. For Webspam, the entries are non-negative and we simply set the positive elements to 1. For Leukemia, we treat each data point as the query and other points as the database for search. For Webspam, we use the training set as the database, and the test set as queries. For each query point, we set the ground truth ("gold-standard" ) neighbors as the top 50 data points in the database with the highest Jaccard similarities to the query.

**Setup.** To search with DP-OPH and DP-MH, we generate the private hash values and compute the collision estimator between the query and each data point. Then, we retrieve the data points with the highest estimated Jaccard similarities to the query. For densified DP-OPH (Algorithm 4) and DP-MH (Algorithm 6), we ensure the lower bound $f$ on the number of non-zero elements by filtering the data points with at least $f$ non-zeros. We use $f = 1000, 500$ for Leukemia and Webspam, respectively, which cover $100\%$ and $90\%$ of the total data points.

**Results.** In Figure 3, we report the precision for Leukemia with $b = 1, 2, 4$ and $\epsilon \in [1, 30]$. The $\epsilon$ range is common in the literature of DP hashing, e.g., the $[2.45, 33.5]$ reported in Zhao et al. (2022) which studied private count-sketch. The recall comparisons are similar. The results are averaged over all query points and over 5 runs. We observe that:

- DP-OPH-re outperforms DP-MH and DP-OPH-fix, at all $\epsilon$ levels. That is, DP-OPH-re is a uniformly more superior method than the existing DP-MH method for private hashing.

- DP-OPH-rand achieves good accuracy with small $\epsilon$ (e.g., $\epsilon < 5$), but stops improving with $\epsilon$ afterwards (due to the random bits for the empty bins), justifying the trade-off discussed in Section 3.2. When $\epsilon$ gets larger (e.g., $\epsilon = 5 \sim 15$), DP-OPH-re performs the best.

The results on Webspam are presented in Figure 4. Similarly, DP-OPH-re achieves better performance than DP-MH and DP-OPH-fix for all $\epsilon$. DP-OPH-rand performs the best with $\epsilon < 10$. DP-OPH-re bypasses DP-OPH-rand as $\epsilon$ grows larger.

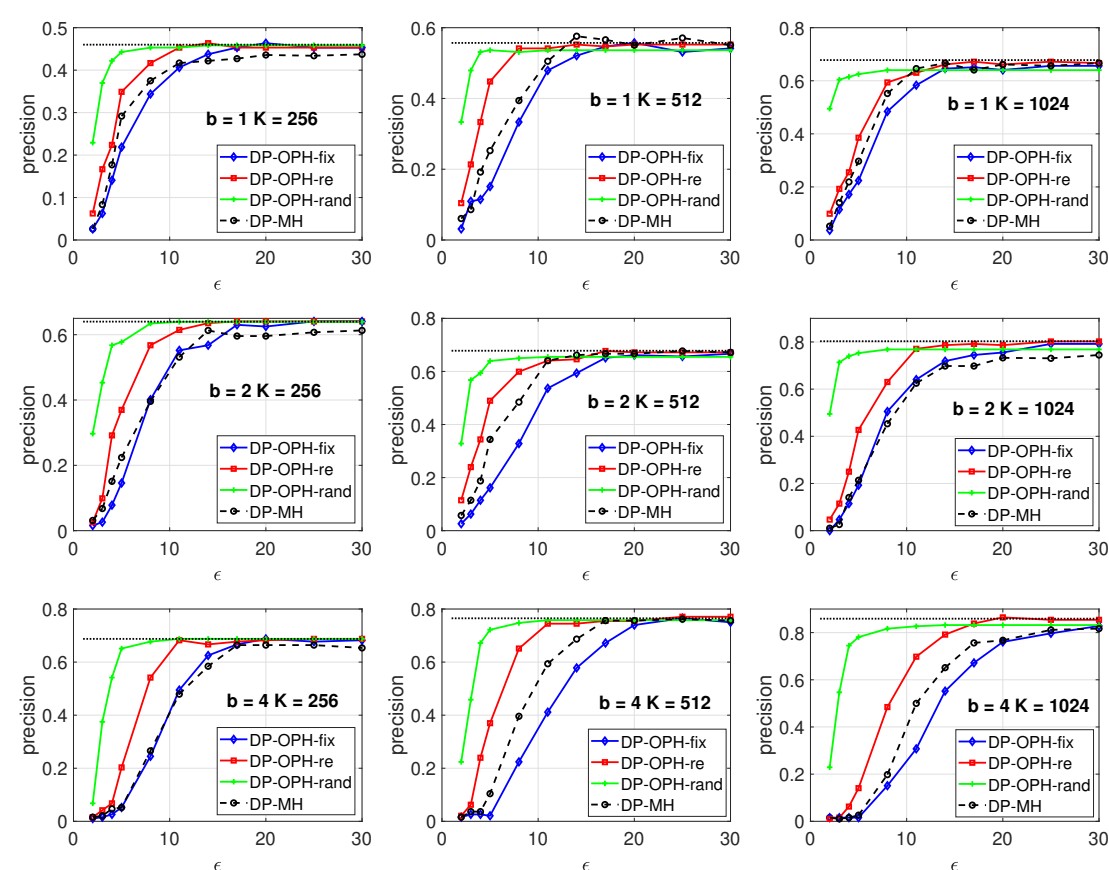

Figure 3: Precision@1 results on Leukemia gene expression dataset with $b = 1, 2, 4$. $\delta = 10^{-6}$. Dotted curves are for non-private OPH-re.

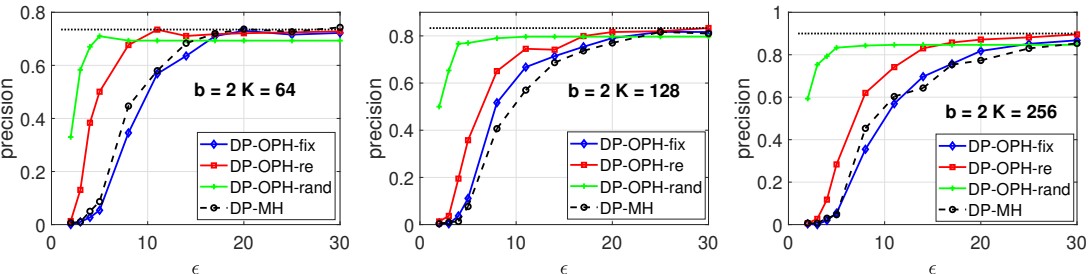

Figure 4: Precision@10 results on Webspam dataset with $b = 2$. $\delta = 10^{-6}$.

## 5 CONCLUSION

In this paper, we study differentially privatized one permutation hashing (DP-OPH) methods. We develop three variants depending on the densification procedure of OPH, and provide privacy and utility analyses of our algorithms. We show the significant advantages of our DP-OPH over the DP MinHash alternative proposed in prior literature for hashing the Jaccard similarity at various privacy levels. Experiments are conducted on retrieval tasks to justify the effectiveness of the proposed DP-OPH, and provide guidance on the appropriate choice of the DP-OPH variant in different scenarios. In Appendix A, we also provide DP-BCWS which is based on bin-wise consistent weighted samples (BCWS) (Li et al., 2019) for weighted Jaccard similarity (for non-negative data). Given the efficiency and good performance, we expect DP-OPH to serve as a useful privatized alternative in practical applications where MinHash-type methods are heavily used. In the appendix, we also provide an extension of DP-OPH to real-value data called DP-BCWS.

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

# A EXTENSION: DIFFERENTIALLY PRIVATE BIN-WISE CONSISTENT WEIGHTED SAMPLING (DP-BCWS) FOR WEIGHTED JACCARD SIMILARITY

---
**Algorithm 7** Consistent Weighted Sampling (CWS)

---
**Input:** Non-negative data vector $\boldsymbol{u} \in \mathbb{R}_+^D$
**Output:** Consistent weighted sampling hash $h^* = (i^*, t^*)$

1: **for** every non-zero $v_i$ **do**
2:      $r_i \sim Gamma(2,1), \ c_i \sim Gamma(2,1), \beta_i \sim Uniform(0,1)$
3:      $t_i \leftarrow \lfloor \frac{\log u_i}{r_i} + \beta_i \rfloor, \ y_i \leftarrow \exp(r_i(t_i - \beta_i))$
4:      $a_i \leftarrow c_i/(y_i \exp(r_i))$
5: **end for**
6: $i^* \leftarrow arg\min_i \ a_i, \qquad t^* \leftarrow t_{i^*}$

---

In our main paper, we focused on DP hashing algorithms for the binary Jaccard similarity. Indeed, our algorithm can also be extended to hashing the weighted Jaccard similarity: (recall the definition)

$$J_w(\boldsymbol{u}, \boldsymbol{v}) = \frac{\sum_{i=1}^D \min\{u_i, v_i\}}{\sum_{i=1}^D \max\{u_i, v_i\}}, \tag{5}$$

for two non-negative data vectors $\boldsymbol{u}, \boldsymbol{v} \in \mathbb{R}_+$. The standard hashing algorithm for (5) is called Consistent Weighted Sampling (CWS) as summarized in Algorithm 7 (Ioffe, 2010; Manasse et al., 2010; Li et al., 2021). To generate one hash value, we need three length-$D$ random vectors $\boldsymbol{r} \sim Gamma(2,1)$, $\boldsymbol{c} \sim Gamma(2,1)$ and $\boldsymbol{\beta} \sim Uniform(0,1)$. We denote Algorithm 7 as a function $CWS(\boldsymbol{u}; \boldsymbol{r}, \boldsymbol{c}, \boldsymbol{\beta})$. Li et al. (2019) proposed bin-wise CWS (BCWS) which exploits the same idea of binning as in OPH. The binning and densification procedure of BCWS is exactly the same as OPH (Algorithm 2 and Algorithm 3), except that every time we apply CWS, instead of MinHash, to the data in the bins to generate hash values. Note that in CWS, the output contains two values: $i^*$ is a location index similar to the output of OPH, and $t^*$ is a real-value scalar. Prior studies (e.g., Li et al. (2021)) showed that the second element has minimal impact on the estimation accuracy in most practical cases (i.e., only counting the collision of the first element suffices). Therefore, in our study, we only keep the first integer element as the hash output for subsequent learning tasks.

For weighted data vectors, we follow the prior DP literature on weighted sets (e.g., Xu et al. (2013); Smith et al. (2020); Dickens et al. (2022); Zhao et al. (2022)) and define the neighboring data vectors as those who differ in one element. To privatize BCWS, there are also three possible ways depending on the densification option. Since the DP algorithm design for densified BCWS requires rigorous and non-trivial computations which might be an independent study, here we empirically test the ($b$-bit) DP-BCWS method with random bits for empty bins. The details are provided in Algorithm 8. In general, we first randomly split the data entries into $K$ equal length bins, and apply CWS to the data $\boldsymbol{u}_{\mathcal{B}_k}$ in each non-empty bin $\mathcal{B}_k$ using the random numbers $(\boldsymbol{r}_{\mathcal{B}_k}, \boldsymbol{c}_{\mathcal{B}_k}, \boldsymbol{\beta}_{\mathcal{B}_k})$ to generated $K$ hash values (possibly including empty bins). After each hash is truncated to $b$ bits, we uniformly randomly assign a hash value in $\{0, ..., 2^b - 1\}$ to every empty bin.

Using the same proof arguments as Theorem 3.3, we have the following guarantee.

**Theorem A.1.** *Algorithm 8 satisfies $\epsilon$-DP.*

**Empirical evaluation.** In Figure 5, we train an $l_2$-regularized logistic regression on the DailySports dataset[2]. and report the test accuracy with various $b$ and $K$ values. The $l_2$ regularization parameter $\lambda$ is tuned over a fine grid from $10^{-4}$ to 10. Similar to the results in the previous section, the performance of DP-BCWS becomes stable as long as $\epsilon > 5$. Note that, linear logistic regression only gives $\approx 75\%$ accuracy on original DailySports dataset (without DP). With DP-BCWS, the accuracy can reach $\approx 98\%$ with $K = 1024$ and $\epsilon = 5$.

---
[2]https://archive.ics.uci.edu/ml/datasets/daily+and+sports+activities

---

**Algorithm 8** Differential Private Bin-wise Consistent Weighted Sampling (DP-BCWS)

---

**Input:** Binary vector $\boldsymbol{u} \in \{0,1\}^D$; number of hash values $K$; number of bits per hash $b$

**Output:** DP-BCWS hash values $\tilde{h}_1(\boldsymbol{u}), ..., \tilde{h}_K(\boldsymbol{u})$

1: Generate length-$D$ random vectors $\boldsymbol{r} \sim Gamma(2,1)$, $\boldsymbol{c} \sim Gamma(2,1)$, $\boldsymbol{\beta} \sim Uniform(0,1)$
2: Let $d = D/K$. Use a permutation $\pi : [D] \mapsto [D]$ with fixed seed to randomly split $[D]$ into $K$ equal-size bins $\mathcal{B}_1, ..., \mathcal{B}_K$, with $\mathcal{B}_k = \{j \in [D] : (k-1)d + 1 \le \pi(j) \le kd\}$
3: **for** $k = 1$ to $K$ **do**
4:     **if** Bin $\mathcal{B}_k$ is non-empty **then**
5:         $h_k(\boldsymbol{u}) \leftarrow CWS(\boldsymbol{u}_{\mathcal{B}_k}; \boldsymbol{r}_{\mathcal{B}_k}, \boldsymbol{c}_{\mathcal{B}_k}, \boldsymbol{\beta}_{\mathcal{B}_k})$         ▷ Run CWS within each non-empty bin
6:         $h_k(\boldsymbol{u}) \leftarrow$ last $b$ bits of $h_k(\boldsymbol{u})$

7:         $\tilde{h}_k(\boldsymbol{u}) = \begin{cases} h_k(\boldsymbol{u}), & \text{with probability } \frac{e^\epsilon}{e^\epsilon + 2^b - 1} \\ i, & \text{with probability } \frac{1}{e^{\epsilon'} + 2^b - 1}, \end{cases}$ for $i \in \{0, ..., 2^b - 1\}$, $i \ne h_k(\boldsymbol{u})$

8:     **else**
9:         $h_k(\boldsymbol{u}) \leftarrow E$
10:         $\tilde{h}_k(\boldsymbol{u}) = i$ with probability $\frac{1}{2^b}$, for $i = 0, ..., 2^b - 1$   ▷ Assign random bits to empty bin
11:     **end if**
12: **end for**

---

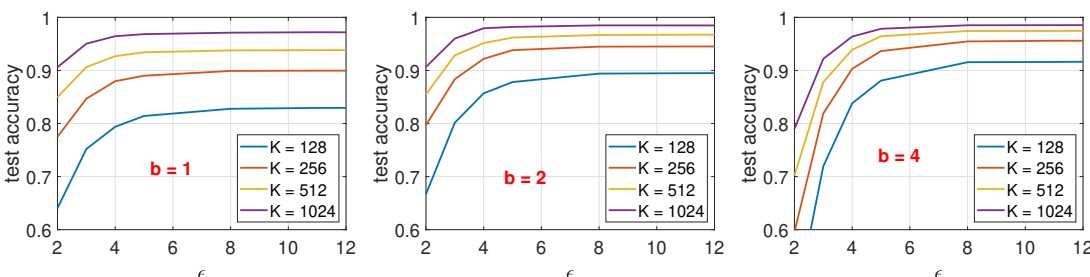

Figure 5: Test classification accuracy of DP-BCWS on DailySports dataset (Asuncion & Newman, 2007) with $l_2$-regularized logistic regression.

In Figure 6, we train a neural network with two hidden layers of size 256 and 128 respectively on MNIST. We use the ReLU activation function and the standard cross-entropy loss. We see that, in a reasonable privacy regime (e.g., $\epsilon < 10$), DP-BCWS is able to achieve $\approx 95\%$ test accuracy with proper $K$ and $b$ combinations (one can choose the values depending on practical scenarios and needs). For example, with $b = 4$ and $K = 128$, DP-BCWS achieves $\approx 97\%$ accuracy at $\epsilon = 8$.

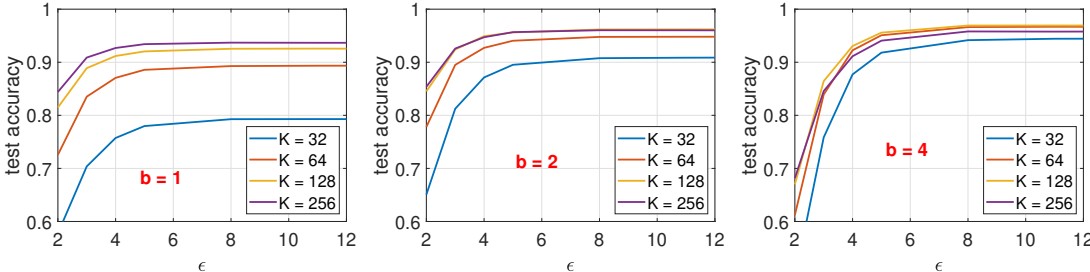

Figure 6: Test classification accuracy of DP-BCWS on MNIST with 2-hidden layer neural network.

## B PROOFS

**Lemma B.1** (Li et al. (2012)). *Let* $f = |\{i : u_i = 1\}|$, *and* $I_{emp,k}$ *be the indicator function that the* $k$-*th bin is empty, and* $N_{emp} = \sum_{k=1}^{K} I_{emp,k}$. *Suppose* $mod(D, K) = 0$. *We have*

$$P\left(N_{emp} = j\right) = \sum_{\ell=0}^{K-j} (-1)^\ell \binom{K}{j} \binom{K-j}{\ell} \binom{D(1-(j+\ell)/K)}{f} \Big/ \binom{D}{f}.$$

**Lemma B.2** (Li et al. (2019)). *Conditional on the event that* $m$ *bins are non-empty, let* $\tilde{f}$ *be the number of non-zero elements in a non-empty bin. Denote* $d = D/K$. *The conditional probability distribution of* $\tilde{f}$ *is given by*

$$P\left(\tilde{f} = j\big|m\right) = \frac{\binom{d}{j} H(m-1, f-j|d)}{H(m, f|d)}, \quad j = \max\{1, f-(m-1)d\}, ..., \min\{d, f-m+1\},$$

*where* $H(\cdot)$ *follows the recursion: for any* $0 < k \leq K$ *and* $0 \leq n \leq f$,

$$H(k, n|d) = \sum_{i=\max\{1, n-(k-1)d\}}^{\min\{d, n-k+1\}} \binom{d}{i} H(k-1, n-i|d), \quad H(1, n|d) = \binom{d}{n}.$$

### B.1 PROOF OF LEMMA 3.1

*Proof.* Without loss of generality, suppose $\boldsymbol{u}$ and $\boldsymbol{u}'$ differ in the $i$-th dimension, and by the symmetry of DP, we can assume that $u_i = 1$ and $u_i' = 0$. We know that $i$ is assigned to the $\lceil mod(\pi(i), d) \rceil$-th bin. Among the $K$ hash values, this change will affect all the bins that uses the data/hash of the $k^* = \lceil mod(\pi(i), d) \rceil$-th bin (after permutation), both in the first scan (if it is non-empty) and in the densification process. Let $N_{emp}$ be the number of empty bins in $h(\boldsymbol{u})$, and $\tilde{f}$ be the number of non-zero elements in the $k^*$-th bin. We have, for $x = 0, ..., K - \lceil f/d \rceil$,

$$P\left(X = x\right) = \sum_{j=\max(0, K-f)}^{K-\lceil f/d \rceil} \sum_{z=1}^{\min(f, d)} P\left(X = x\big|\tilde{f} = z, N_{emp} = j\right) P\left(\tilde{f} = z, N_{emp} = j\right)$$

$$= \sum_{j=\max(0, K-f)}^{K-\lceil f/d \rceil} \sum_{z=1}^{\min(f, d)} P\left(X = x\big|\tilde{f} = z, N_{emp} = j\right) P\left(\tilde{f} = z|K-j\right) P\left(N_{emp} = j\right),$$

where $P\left(\tilde{f} = z|K-j\right)$ is given in Lemma B.2 and $P\left(N_{emp} = j\right)$ can be calculated by Lemma B.1. To compute the first conditional probability, we need to compute the number of times the $k^*$-th bin is picked to generated hash values, and the hash values are different for $\boldsymbol{u}$ and $\boldsymbol{u}'$. Conditional on $\{\tilde{f} = z, N_{emp} = j\}$, denote $\Omega = \{k : \mathcal{B}_k \text{ is empty}\}$, and let $R_k$ be the non-empty bin used for the $k$-th hash value $h_k(\boldsymbol{u})$, which takes value in $[K] \setminus \Omega$. We know that $|\Omega| = j$. We can write

$$X = \mathbb{1}\{h_{k^*}(\boldsymbol{u}) \neq h_{k^*}(\boldsymbol{u}')\} + \sum_{k \in \Omega} \mathbb{1}\{R_k = k^*, h_k(\boldsymbol{u}) \neq h_k(\boldsymbol{u}')\}.$$

Here we separate out the first term because the $k^*$-th hash always uses the $k^*$-bin. Note that the densification bin selection is uniform, and the bin selection is independent of the permutation for hashing. For the fixed densification, since the hash value $h_{k^*}(\boldsymbol{u})$ is generated and used for all hash values that use $\mathcal{B}_{k^*}$, we have

$$P\left(X = x\big|\tilde{f} = z, N_{emp} = j\right) = \mathbb{1}\{x = 0\}\left(1 - P_{\neq}\right) + \mathbb{1}\{x > 0\} P_{\neq} \cdot g_{bino}\left(x - 1; \frac{1}{K-j}, j\right),$$

where $g_{bino}(x; p, n)$ is the probability mass function of the binomial distribution with $n$ trials and success rate $p$, and $P_{\neq} = P(h_{k^*}(\boldsymbol{u}) \neq h_{k^*}(\boldsymbol{u}')) = \left(1 - \frac{1}{2^b}\right)\frac{1}{z}$. Based on the same reasoning, for re-randomized densification, we have

$$P\left(X = x\big|\tilde{f} = z, N_{emp} = j\right) = (1 - P_{\neq}) \cdot g_{bino}\left(x; \frac{P_{\neq}}{K-j}, j\right) + P_{\neq} \cdot g_{bino}\left(x - 1; \frac{P_{\neq}}{K-j}, j\right).$$

Combining all the parts together completes the proof. □

## B.2 PROOF OF THEOREM 3.2

*Proof.* Let $\boldsymbol{u}$ and $\boldsymbol{u}'$ be neighbors only differing in one element. Denote $S = \{k \in [K] : h_k(\boldsymbol{u}) \neq h_k(\boldsymbol{u}')\}$ and $S^c = [K] \setminus S$. As discussed before, we can verify that for $k \in S_c$, we have $\frac{P(\tilde{h}_k(\boldsymbol{u})=i)}{P(\tilde{h}_k(\boldsymbol{u}')=i)} = 1$ for any $i = 0, ..., 2^b - 1$. For $k \in S$, $e^{-\epsilon'} \leq \frac{P(\tilde{h}_k(\boldsymbol{u})=i)}{P(\tilde{h}_k(\boldsymbol{u}')=i)} \leq e^{\epsilon'}$ holds for any $i = 0, ..., 2^b - 1$. Thus, for any $Z \in \{0, ..., 2^b - 1\}^K$, the absolute privacy loss can be bounded by

$$\left| \log \frac{P(\tilde{h}(\boldsymbol{u}) = Z)}{P(\tilde{h}(\boldsymbol{u}') = Z)} \right| = \left| \log \prod_{k \in S} \frac{P(\tilde{h}_k(\boldsymbol{u}) = i)}{P(\tilde{h}_k(\boldsymbol{u}') = i)} \right| \leq |S|\epsilon' = |S|\frac{\epsilon}{N}. \tag{6}$$

By Lemma 3.1, with probability $1-\delta$, $|S| \leq F_{fix}^{-1}(1-\delta) = N$ for DP-OPH-fix; $|S| \leq F_{re}^{-1}(1-\delta) = N$ for DP-OPH-re. Hence, (6) is bounded by $\epsilon$ with probability $1-\delta$. This proves the $(\epsilon, \delta)$-DP. $\square$

## B.3 PROOF OF THEOREM 3.3

*Proof.* The proof is similar to the proof of Theorem 3.2. Since the original hash vector $h(\boldsymbol{u})$ is not densified, there only exists exactly one hash value such that $h_k(\boldsymbol{u}) \neq h_k(\boldsymbol{u})$ may happen for $\boldsymbol{u}'$ that differs in one element from $\boldsymbol{u}$. W.l.o.g., assume $u_i = 1$ and $u_i' = 0$, and $i \in \mathcal{B}_k$. If bin $k$ is non-empty for both $\boldsymbol{u}$ and $\boldsymbol{u}'$ (after permutation), then for any $Z \in \{0, ..., 2^b - 1\}^K$, $\left| \log \frac{P(\tilde{h}(\boldsymbol{u})=Z)}{P(\tilde{h}(\boldsymbol{u}')=Z)} \right| \leq \epsilon$ according to our analysis in Theorem 3.2 (the probability of hash in $[K] \setminus \{k\}$ cancels out). If bin $k$ is empty for $\boldsymbol{u}'$, since $1 \leq \frac{e^\epsilon}{e^\epsilon+2^b-1} / \frac{1}{2^b} \leq e^\epsilon$ and $e^{-\epsilon} \leq \frac{1}{2^b} / \frac{1}{e^\epsilon+2^b-1} \leq 1$, we also have $\left| \log \frac{P(\tilde{h}(\boldsymbol{u})=Z)}{P(\tilde{h}(\boldsymbol{u}')=Z)} \right| \leq \epsilon$. Therefore, the algorithm is $\epsilon$-DP as claimed. $\square$

## B.4 PROOF OF THEOREM 3.5

*Proof.* For the two densified DP-OPH variants, DP-OPH-fix and DP-OPH-re, and the DP MinHash (DP-MH) methods, each full-precision (and unprivatized) hash value of $h(\boldsymbol{u})$ and $h(\boldsymbol{v})$ has collision probability equal to $P(h(\boldsymbol{u}) = h(\boldsymbol{v})) = J(\boldsymbol{u}, \boldsymbol{v})$. Let $h^{(b)}(\boldsymbol{u})$ denote the $b$-bit hash values. Since we assume the last $b$ bits are uniformly assigned, we have $P(h^{(b)}(\boldsymbol{u}) = h^{(b)}(\boldsymbol{v})) = J + (1-J)\frac{1}{2^b}$. Denote $p = \frac{\exp(\epsilon/N)}{\exp(\epsilon/N)+2^b-1}$. By simple probability calculation, the privatized $b$-bit hash values has collision probability

$$P(\tilde{h}(\boldsymbol{u}) = \tilde{h}(\boldsymbol{v}))$$

$$= P(\tilde{h}(\boldsymbol{u}) = \tilde{h}(\boldsymbol{v})|h^{(b)}(\boldsymbol{u}) = h^{(b)}(\boldsymbol{v}))P(h^{(b)}(\boldsymbol{u}) = h^{(b)}(\boldsymbol{v}))$$

$$+ P(\tilde{h}(\boldsymbol{u}) = \tilde{h}(\boldsymbol{v})|h^{(b)}(\boldsymbol{u}) \neq h^{(b)}(\boldsymbol{v}))P(h^{(b)}(\boldsymbol{u}) \neq h^{(b)}(\boldsymbol{v}))$$

$$= \left[ p^2 + \frac{(1-p)^2}{2^b - 1} \right] \left( \frac{1}{2^b} + \frac{2^b - 1}{2^b}J \right) + \left[ \frac{2p(1-p)}{2^b - 1} + \frac{2^b - 2}{(2^b - 1)^2}(1-p)^2 \right] \left( \frac{2^b - 1}{2^b} - \frac{2^b - 1}{2^b}J \right)$$

$$= \left[ p^2 + \frac{(1-p)^2}{2^b - 1} - \frac{2p(1-p)}{2^b - 1} - \frac{2^b - 2}{(2^b - 1)^2}(1-p)^2 \right] \frac{2^b - 1}{2^b}J$$

$$+ \frac{1}{2^b}\left[ p^2 + \frac{(1-p)^2}{2^b - 1} + 2p(1-p) + \frac{2^b - 2}{2^b - 1}(1-p)^2 \right]$$

$$= \left[ p^2 + \frac{(1-p)^2 - 2(2^b - 1)p(1-p)}{(2^b - 1)^2} \right] \frac{2^b - 1}{2^b}J + \frac{1}{2^b}\left[ p^2 + 2p(1-p) + (1-p)^2 \right]$$

$$= \frac{(2^b p + 1)^2}{2^b(2^b - 1)}J + \frac{1}{2^b},$$

which implies $J = \frac{(2^b - 1)(2^b P(\tilde{h}(u)=\tilde{h}(v))-1)}{(2^b p - 1)^2}$. Therefore, let $\hat{J} = \frac{1}{K}\sum_{k=1}^K \mathbb{1}\{\tilde{h}_k(u) = \tilde{h}_k(v)\}$, then an unbiased estimator of $J$ can be formulated as

$$\hat{J}_{unbias} = \frac{(2^b - 1)(2^b \hat{J} - 1)}{(2^b p - 1)^2}.$$

$\square$

### B.5 PROOF OF THEOREM 3.6

*Proof.* As before, define $\hat{J} = \frac{1}{K}\sum_{k=1}^{K}\mathbb{1}\{\tilde{h}_k(u) = \tilde{h}_k(v)\}$. For all three methods, we know that $\mathbb{E}[\hat{J}] = \frac{(2^b p+1)^2}{2^b(2^b-1)}J + \frac{1}{2^b}$. Denote $J_B = P(h^{(b)}(\boldsymbol{u}) = h^{(b)}(\boldsymbol{v})) = J + (1-J)\frac{1}{2^b}$. $\hat{J}_{unbias} = \frac{(2^b-1)(2^b\hat{J}-1)}{(2^b p-1)^2}$.

**MinHash.** We have

$$Var[\hat{J}] = \mathbb{E}[\hat{J}^2] - \mathbb{E}[\hat{J}]^2$$

$$= \frac{1}{K^2}\mathbb{E}\left[\sum_{i=1}^{K}\mathbb{1}\{\tilde{h}_i(u) = \tilde{h}_i(v)\}] + \sum_{i\neq j}\mathbb{1}\{\tilde{h}_i(u) = \tilde{h}_i(v)\}\mathbb{1}\{\tilde{h}_j(u) = \tilde{h}_j(v)\}\right] - \left(\frac{(2^b p+1)^2}{2^b(2^b-1)}J + \frac{1}{2^b}\right)^2$$

$$= \frac{1}{K}\left(\frac{(2^b p+1)^2}{2^b(2^b-1)}J + \frac{1}{2^b}\right) + \frac{K-1}{K}A - \left(\frac{(2^b p+1)^2}{2^b(2^b-1)}J + \frac{1}{2^b}\right)^2,$$

where $A = \mathbb{E}[\mathbb{1}\{\tilde{h}_i(u) = \tilde{h}_i(v), \tilde{h}_j(u) = \tilde{h}_j(v)\}]$ for $i \neq j$. The key is to calculate $A$. By symmetry,

$$A$$

$$= P(\tilde{h}_i(u) = \tilde{h}_i(v), \tilde{h}_j(u) = \tilde{h}_j(v)|h_i^{(b)}(u) = h_i^{(b)}(v), h_j^{(b)}(u) = h_j^{(b)}(v))P(h_i^{(b)}(u) = h_i^{(b)}(v), h_j^{(b)}(u) = h_j^{(b)}(v))$$

$$+ 2P(\tilde{h}_i(u) = \tilde{h}_i(v), \tilde{h}_j(u) = \tilde{h}_j(v)|h_i^{(b)}(u) = h_i^{(b)}(v), h_j^{(b)}(u) \neq h_j^{(b)}(v))P(h_i^{(b)}(u) = h_i^{(b)}(v), h_j^{(b)}(u) \neq h_j^{(b)}(v))$$

$$+ P(\tilde{h}_i(u) = \tilde{h}_i(v), \tilde{h}_j(u) = \tilde{h}_j(v)|h_i^{(b)}(u) \neq h_i^{(b)}(v), h_j^{(b)}(u) \neq h_j^{(b)}(v))P(h_i^{(b)}(u) \neq h_i^{(b)}(v), h_j^{(b)}(u) \neq h_j^{(b)}(v))$$

$$:= A_{11} + 2A_{01} + A_{00}.$$

By independence, we have

$$A_{11} = \left(p^2 + \frac{(1-p)^2}{2^b-1}\right)^2 J_B^2$$

$$A_{10} = \left(p^2 + \frac{(1-p)^2}{2^b-1}\right)\left(\frac{2p(1-p)}{2^b-1} + \frac{2^b-2}{(2^b-1)^2}(1-p)^2\right)J_B(1-J_B)$$

$$A_{00} = \left(\frac{2p(1-p)}{2^b-1} + \frac{2^b-2}{(2^b-1)^2}(1-p)^2\right)^2(1-J_B)^2,$$

which leads to

$$A = \left(\frac{(2^b p+1)^2}{2^b(2^b-1)}J + \frac{1}{2^b}\right)^2.$$

Thus, we have

$$Var[\hat{J}] = \frac{1}{K}\left(\frac{(2^b p+1)^2}{2^b(2^b-1)}J + \frac{1}{2^b}\right)\left(\frac{2^b-1}{2^b} - \frac{(2^b p+1)^2}{2^b(2^b-1)}J\right)$$

and

$$Var[\hat{J}_{unbias,MH}] = \frac{2^{2b}(2^b-1)^2}{(2^b p-1)^4}Var[\hat{J}]$$

$$= \frac{1}{K}\left(\frac{(2^b p+1)^2}{(2^b p-1)^2}J + \frac{2^b-1}{(2^b p-1)^2}\right)\left(\frac{(2^b-1)^2}{(2^b p-1)^2} - \frac{(2^b p+1)^2}{(2^b p-1)^2}J\right).$$

**DP-OPH-fix.** We write $\hat{J} = \frac{1}{K}\sum_{k=1}^{K}(\tilde{I}_k^N + \tilde{I}_k^E)$, where $\tilde{I}_k^N$ is the indicator function of hash collision at the $k$-th bin and when the bin is non-empty, and $\tilde{I}_k^N$ is the indicator function of hash

collision at the $k$-th bin and when the bin is empty. Similar to previous analysis,

$$Var[\hat{J}] = \frac{1}{K^2} \mathbb{E}\left[ (\sum_{k=1}^{K} (\tilde{I}_k^N + \tilde{I}_k^E))^2 \right] - \left( \frac{(2^b p + 1)^2}{2^b(2^b - 1)} J + \frac{1}{2^b} \right)^2$$

$$= \frac{1}{K} \left( \frac{(2^b p + 1)^2}{2^b(2^b - 1)} J + \frac{1}{2^b} \right) + \frac{1}{K^2} A - \left( \frac{(2^b p + 1)^2}{2^b(2^b - 1)} J + \frac{1}{2^b} \right)^2,$$

where

$$A = \mathbb{E}[\sum_{i \neq j} (\tilde{I}_i^N + \tilde{I}_i^E)(\tilde{I}_j^N + \tilde{I}_j^E)]$$

$$= \mathbb{E}_m \left[ \mathbb{E}[m(m-1)\tilde{I}_i^N \tilde{I}_j^N + 2m(K-m)\tilde{I}_i^N \tilde{I}_j^E + (K-m)(K-m-1)\tilde{I}_i^E \tilde{I}_j^E]|m] \right]. \quad (7)$$

Here the condition on "$\cdot|m$" means the event that there are $m$ simultaneously non-empty bins. Denote $I_k = \mathbb{1}\{h_k(u) = h_k(v)\}$ be the collision indicator of the original hash values, and $I_k^{(b)} = \mathbb{1}\{h_k^{(b)}(u) = h_k^{(b)}(v)\}$ be the collision indicator of the $b$-bit hash values. For two non-empty bins $i$ and $j$, we have

$$\tau_{11} := P(h_i(u) = h_i(v), h_j(u) = h_j(v)|m) = \mathbb{E}[I_i I_j | m] = J\tilde{J},$$

$$\tau_{10} := P(h_i(u) = h_i(v), h_j(u) \neq h_j(v)|m) = \mathbb{E}[I_i(1 - I_j)|m] = J - J\tilde{J},$$

$$\tau_{00} := P(h_i(u) \neq h_i(v), h_j(u) \neq h_j(v)|m) = \mathbb{E}[(1 - I_i)(1 - I_j)|m] = 1 - 2J + J\tilde{J},$$

and using total probability formula (conditional on $h_i$ and $h_j$),

$$P(h_i^{(b)}(u) = h_i^{(b)}(v), h_j^{(b)}(u) = h_j^{(b)}(v)|m) = \mathbb{E}[I_i^{(b)} I_j^{(b)}|m]$$

$$= \tau_{11} + 2\frac{1}{2^b}\tau_{10} + \frac{1}{2^{2b}}\tau_{00}$$

$$= J\tilde{J} + \frac{1}{2^{b-1}}(J - J\tilde{J}) + \frac{1}{2^{2b}}(1 - 2J + J\tilde{J}) := P_{11}$$

$$P(h_i^{(b)}(u) = h_i^{(b)}(v), h_j^{(b)}(u) \neq h_j^{(b)}(v)|m) = \mathbb{E}[I_i^{(b)}(1 - I_j^{(b)})|m] = J_B - P_{11} := P_{10}$$

$$P(h_i^{(b)}(u) \neq h_i^{(b)}(v), h_j^{(b)}(u) \neq h_j^{(b)}(v)|m) = \mathbb{E}[(1 - I_i^{(b)})(1 - I_j^{(b)})|m] = 1 - 2J_B + P_{11} := P_{00}.$$

Thus,

$$\mathbb{E}[\tilde{I}_i^N \tilde{I}_j^N | m] = \left( p^2 + \frac{(1-p)^2}{2^b - 1} \right)^2 P_{11} + 2 \left( p^2 + \frac{(1-p)^2}{2^b - 1} \right) \left( \frac{2p(1-p)}{2^b - 1} + \frac{2^b - 2}{(2^b - 1)^2}(1-p)^2 \right) P_{10}$$

$$+ \left( \frac{2p(1-p)}{2^b - 1} + \frac{2^b - 2}{(2^b - 1)^2}(1-p)^2 \right)^2 P_{00}.$$

For two empty bins $i$ and $j$, we have, for fixed densification,

$$\tau_{11,f} = P(h_i(u) = h_i(v), h_j(u) = h_j(v)|m) = \frac{1}{m}J + \frac{m-1}{m}J\tilde{J},$$

$$\tau_{10,f} = P(h_i(u) = h_i(v), h_j(u) \neq h_j(v)|m) = \mathbb{E}[I_i(1 - I_j)] = \frac{m-1}{m}(J - J\tilde{J}),$$

$$\tau_{00,f} = P(h_i(u) \neq h_i(v), h_j(u) \neq h_j(v)|m) = \mathbb{E}[(1 - I_i)(1 - I_j)] = 1 - (2 - \frac{1}{m})J + \frac{m-1}{m}J\tilde{J}.$$

Similarly,

$$\mathbb{E}[\tilde{I}_i^E \tilde{I}_j^E | m] = \left( p^2 + \frac{(1-p)^2}{2^b - 1} \right)^2 P_{11,f} + 2 \left( p^2 + \frac{(1-p)^2}{2^b - 1} \right) \left( \frac{2p(1-p)}{2^b - 1} + \frac{2^b - 2}{(2^b - 1)^2}(1-p)^2 \right) P_{10,f}$$

$$+ \left( \frac{2p(1-p)}{2^b - 1} + \frac{2^b - 2}{(2^b - 1)^2}(1-p)^2 \right)^2 P_{00,f},$$

where

$$P_{11,f} = \tau_{11,f} + \frac{1}{2^{b-1}}\tau_{10,f} + \frac{1}{2^{2b}}\tau_{00,f}, \quad P_{10,f} = J_B - P_{11,f}, \quad P_{00,f} = 1 - 2J_B + P_{11,f}.$$

It is not hard to note that $\mathbb{E}[\tilde{I}_i^N \tilde{I}_j^E | m] = \mathbb{E}[\tilde{I}_i^E \tilde{I}_j^E | m]$. Putting pieces together into (7), we get the variance for DP-OPH-fix.

**DP-OPH-re.** For DP-OPH-re, most calculations are the same as DP-OPH-fix. According to Li et al. (2019), we have

$$\tau_{11,r} = P(h_i(u) = h_i(v), h_j(u) = h_j(v)|m) = \mathbb{E}[I_i I_j] = \frac{\zeta(m)}{m}J + \frac{m - \zeta(m)}{m}J\tilde{J},$$

$$\tau_{10,r} = P(h_i(u) = h_i(v), h_j(u) \neq h_j(v)|m) = \mathbb{E}[I_i(1 - I_j)] = \frac{m - \zeta(m)}{m}(J - J\tilde{J}),$$

$$\tau_{00,r} = P(h_i(u) \neq h_i(v), h_j(u) \neq h_j(v)|m) = \mathbb{E}[(1 - I_i)(1 - I_j)]$$
$$= 1 - (2 - \frac{\zeta(m)}{m})J + \frac{m - \zeta(m)}{m}J\tilde{J},$$

with $\zeta(m) = \mathbb{E}[\frac{1}{\tilde{f}}|m]$ where the conditional distribution of $\tilde{f}$ is given in Lemma B.2. We then get $P_{11,r}$, $P_{10,r}$, $P_{00,r}$ correspondingly. Plugging them into the formula above completes the proof. $\square$

