# OpenReview forum: "Differentially Private One Permutation Hashing"
_ICLR.cc/2026/Conference — Submitted to ICLR 2026_

### Official Review · Reviewer_yXfd · 2025-10-21

**Soundness:** 3
**Presentation:** 3
**Contribution:** 2
**Rating:** 6
**Confidence:** 2

**Summary:**

The paper addresses the privacy of output sketches generated by One Permutation Hashing (OPH) , an efficient algorithm used to approximate Jaccard similarity for large-scale search and learning. The authors propose a framework called DP-OPH, which combines OPH with differential privacy (DP).

**Strengths:**

The strengths of the paper are as follows:
1. The paper's originality stems from its focused investigation into the differential privacy of One Permutation Hashing (OPH) , a topic the authors note has not been well-studied. The primary contribution is the novel DP-OPH framework.
2. The quality of the work is high, demonstrating rigor in both its theoretical and empirical sections.
3. The paper is written with high clarity and is well-structured.

**Weaknesses:**

The weaknesses of the paper are as follows:
1. The paper's experimental solution to this is to ensure the lower bound by filtering the data points. This has two practical drawbacks, i. This approach forced the authors to discard 10% of the Webspam dataset. This is a significant utility loss before any privacy noise is even added. ii. The privacy guarantee is contingent on this pre-filtering. It is unclear how the system would handle new data points that fall below a threshold. A practitioner would have to choose to either discard the new data (losing information) or violate the privacy guarantee for that data point.
2. The authors only develop the DP-BCWS-rand variant (i.e., no densification). The paper's own results on binary data show that the rand variant is suboptimal in many common utility regimes (i.e., when $\epsilon > 5$) compared to densified versions. The authors state that designing densified versions for BCWS is "non-trivial", which effectively punts on the most promising versions of the algorithm.

**Questions:**

Rebuttal questions:
1. The $(\epsilon, \delta)$-DP guarantees for DP-OPH-fix and DP-OPH-re depend on a known lower bound $f$, the minimum number of non-zero elements in any data vector. In the experiments, this is handled by filtering the data, which resulted in discarding 10% of the Webspam dataset. This pre-filtering is a utility loss that is not captured in the subsequent precision/MSE plots. How should a practitioner using this framework handle new, incoming data points that fall below the chosen threshold $f$? Must they be discarded (losing data) or processed (violating the stated $(\epsilon, \delta)$-DP guarantee)?
2. How was the value of $\delta=10^{-6}$ chosen?
3. The authors state that designing densified DP-BCWS is non-trivial. Could you please elaborate on the specific technical challenges that prevent the application of the DP-OPH-fix/re strategies to the weighted case?
4. To make the impact clearer, could the authors briefly comment on the practical consequences? For example, does using the correct bound result in a significantly larger $N$ compared to the flawed calculation in the prior work?

---

> ### Author Response · Authors · 2025-12-04
> **Reply to Reviewer yXfd**
>
> Dear Reviewer yXfd,
>
> Thanks for your valuable feedback on our work and insightful detailed comments.
>
> One of the main questions is the specification of the lower-bound f. Let’s first provide some explanations on the impact of the choice of f.
>
> 1. First of all, in practice, the “f” value can be pre-determined and enforced. This is different from ML experiments on existing data for the purpose of writing papers. For example, if we use n-grams as sparse embeddings for RAG systems, we can easily enforce “f” in the
> implementation. If a sentence is too short, we can combine it with the neighboring sentence. We can also increase $n$ in the n-gram so that we get many more non-zeros. In the experiments with the (existing) Webspam dataset, because the purpose is to verify (and compare) the algorithms at some chosen f values, we need to clean the data a bit before the experiment. This is not the same situation when we have the freedom of developing our own datasets in practice.
>
> 2. Even if the actual f is lower than the specified lower bound, the resultant method is still private but at a different $\epsilon$ which can be calculated for the specific input parameters.
>
> 3. Another choice in practice is that we can set several different thresholds on the $f$ to deal with vectors with different sparsity.
>
> Now let’s try to answer the specific questions. Note that Q1 repeats much of the above statements.
>
> Q1. In our experiments, we used a unified lower bound $f$ on the number of non-zeros entries of simplicity. Indeed, the privacy guarantees in Algorithm 4, 5, and 6 hold for any vector that has more than $f$ non-zeros. Therefore, in principle, we can use different flipping probability for every different vector, and set $f$ in the algorithms exactly equal the number of non-zeros of the vector. In our experiments, we hope to apply the same DP-OPH algorithm (one privacy discount factor $N$) per dataset for a more "general" privacy statement that holds for all vectors naturally.
>
> In addition, computing the flipping probabilities in DP-OPH needs non-negligible calculation with its complicated form in Lemma 3.1. Computing it for every vector in a large dataset is not very efficient. In practice, we can also set several intervals on $f$ each using a different discount factor. For example, we may have three flipping probability values for vectors with non-zeros in $[100, 500), [500, 1000), [1000, \infty]$. With some additional corner case handler, we can essentially cover vectors with any sparsity.
>
> Q2.   $\delta<1/n$ where n is the number of data points is a commonly used heuristics. For example, Dwork and Roth (2014) suggested that rule and so did in many research papers. In general,  $\delta=10^{-6}$ appears to be a common choice. We also tested $\delta=10^{-8}$ and the comparisons results are essentially the same. We are happy to provide these additional results in the (revised) appendix, though the figures will look quite similar.
>
> Q3. We are glad that you are interested in the DP-BCWS variant. For DP-BCWS, computing the "optimal" flipping probability rigorously is quite challenging (i.e., the formula would be extremely complicated) because now the non-zeros all have different weights. For binary data, in the analysis, we can treat the problem as the classic balls-and-bins problem and all the balls are indistinguishable. But if every non-zero has a different weight, the combinatorial analysis becomes much more complicated. Nonetheless, our experimental results show that the random bit approach for DP-BCWS works well.
>
> Q4. Yes, the prior work cited in footnote 1 contained an error. If we use the correct formula, the privacy discount factor $N$ would be $3 \sim 5$ times larger than our proposed analysis for $f=200\sim 1000$. Since we directly provide the optimal solution, we did not include this comparison in the submission. Thank you for your suggestion, we will be happy to add some additional figures to compare these two values. Thanks.
>
> Again, we sincerely appreciate your review of our submission. We hope our rebuttal answers your questions adequately. Thank you.

---

### Official Review · Reviewer_mUNu · 2025-10-22

**Soundness:** 2
**Presentation:** 1
**Contribution:** 2
**Rating:** 2
**Confidence:** 4

**Summary:**

This paper develops differential privacy schemes for popular hashing algorithms with Jaccard similarity: the MinHash and one permutation hashing. For OPH, it provides algorithms for various re-densification approaches, including fixed and randomized variants. The DP mechanism is quite simple: it's the randomized response technique that flips bits with a certain probability. Both privacy and utility are analyzed for various DP-OPH and DP-MinHash.

**Strengths:**

The main strength of this paper is to fill in the gap between DP and hashing for Jaccard similarity, in particular by fixing an issue of an early paper for MinHash. The evaluation is quite comprehensive, with various metrics such as sparsity v.s. the privacy parameter of DP v.s. MSE, precision etc. The theoretical claims are sound and proofs are provided and they seem correct to me.

**Weaknesses:**

The DP techniques used are straightforward, the analysis is also standard, so it does not bring new many insights into the scene. The proof boils down to calculating the quality of estimator under randomized response technique. It is quite thin on the front of technical novelty and significance. Maybe one axis that could be improved is to further reduce the variance of the estimator. A minor weakness is on the presentation, e.g., Theorem 3.6 is presented in a very cumbersome way, I think it's fine to present the full statement in the appendix, but in the main body of the paper this theorem contains too many notations and quantities and is very hard to parse. Authors could consider to use a simplified/condensed version in the main body.

**Questions:**

For Theorem 3.6, it seems the variances do not depend on at all? Could you comment on how parameters of DP affect the variances?

---

> ### Author Response · Authors · 2025-12-04
> **Reply to Reviwer mUNu**
>
> Dear Reviewer mUNu:
>
> Thank you very much for your thoughtful review. We sincerely appreciate your positive assessment of our submission. In particular, you wrote:
>
> > *“The main strength of this paper is to fill in the gap between DP and hashing for Jaccard similarity, in particular by fixing an issue of an early paper for MinHash. The evaluation is quite comprehensive… The theoretical claims are sound and proofs are provided and they seem correct to me.”*
>
> This aligns well with our intention: the paper addresses a practically important and long-standing problem, provides new DP hashing schemes, and offers thorough evaluation and sound theory.
>
> Given this, we were a bit puzzled by the final score (2) and by certain listed weaknesses, as they appear inconsistent with the strengths you highlighted.
>
> ---
>
> ### **1. “The DP techniques used are straightforward.”**
>
> We interpret “straightforward” as a positive property of the proposed algorithms. Indeed, many widely influential algorithms (e.g., MinHash, boosting, tree models, CountSketch) are successful precisely because they are simple and practical.
>
> Our goal was to design DP mechanisms **that are simple enough to be deployable** while **effectively addressing a real industry challenge** for Jaccard-similarity–based hashing and non-binary extentions.
>
> The extensive empirical results demonstrate that the proposed methods **do meaningfully solve the practical problem**.
>
> ---
>
> ### **2. “The analysis is also standard.”**
>
> This comment seems inconsistent with the earlier positive assessment:
>
> > *“The theoretical claims are sound and proofs are provided and they seem correct to me.”*
>
> If the proofs are correct, complete, and match the contributions, then whether some steps follow standard probabilistic inequalities (e.g., Chernoff/Hoeffding) does not diminish the contribution—this is the norm in theoretical ML/TCS.
> In fact, most theoretical analysis of widely used algorithms ultimately relies on standard tools.
>
> Thus, we are unsure whether “standard” is intended as a criticism or simply a description of the analysis style.
>
> ---
>
> ### **3. Clarifying the evaluation of significance**
>
> We respectfully hope that the evaluation and score can be aligned with the factual contributions:
>
> * **Significance**: The paper solves a well-known and practically important problem of achieving DP for Jaccard hashing (MinHash/OPH).
> * **Novelty**: We fix a key issue in the classic MinHash-based DP method and propose new mechanisms.
> * **Thoroughness**: We provide extensive experiments across sparsity, privacy parameters, MSE, precision, etc.
> * **Correctness**: You explicitly confirmed that the theoretical claims and proofs appear correct.
>
> These points match the ICLR criteria for significance and quality.
>
> ---
>
> Again, we genuinely appreciate Reviewer's positive comments on the contribution and technical correctness. Since the surfaced weaknesses appear to be based more on subjective impressions than on concrete issues, we do not have substantial corrections to make in the rebuttal.
>
> We respectfully ask that other Reviewers and the AC consider the alignment between the stated strengths and the numerical score.
>
> Thank you again for your review.

---

### Official Review · Reviewer_UvV5 · 2025-10-29

**Soundness:** 3
**Presentation:** 4
**Contribution:** 3
**Rating:** 8
**Confidence:** 3

**Summary:**

The paper presents a concise review of sketching methods for estimating Jaccard similarity, including MinHash, one-permutation hashing, and two of its variants.

It then introduces differentially private versions of these sketches, all based on randomized response techniques.

An experimental study is conducted to evaluate and compare their practical performance.

**Strengths:**

1. The paper is remarkably well written. It presents one-permutation hashing—including its motivation, relevant literature, and underlying techniques—in a clear and well-organized manner.

2. It provides a solid mini-survey of differentially private variants of one-permutation hashing and includes a comparative experimental evaluation of their performance.

**Weaknesses:**

1. From the experimental results, the simple DP-OPH-rand algorithm appears to achieve the best performance for a reasonable range of $\epsilon$. Even for very large $\epsilon$, its performance remains comparable to that of the more complex variants.


2. It might be clearer to present DP-OPH-rand (Algorithm 5) before Algorithm 4. In addition, the unbiased estimator and its variance analysis for Algorithm 5 seem to be missing and would strengthen the presentation if included.


     Since Sections 2 and 3.1 follow the order *MinHash → OPH → OPH-fix/OPH-re*, it would improve consistency and readability if Section 3.2 adopted the same order when introducing the corresponding privatized variants.

2. The proposed algorithms are largely direct combinations of existing techniques.

**Questions:**

Some minor comments on the writing:

1. The unbiased estimator for the hashes seems to be an important part of the algorithms’ interfaces. However, they are not included in the pseudocode. For instance, the estimators for Algorithms 4 and 6 are only presented later, in Theorem 3.5 on page 7.

2. The variance analysis in Theorem 3.6 is somewhat difficult to interpret. First, it relies on definitions provided only in the appendix. Second, it lacks textual explanation or comparison of the bounds. As a result, it is not entirely clear why these results are included in the main body of the paper.

---

> ### Author Response · Authors · 2025-12-04
> **Reply to Reviewer UvV5**
>
> Dear Reviewer UvV5,
>
> Thank you for your support and your kind suggestions on adjusting presentations to improve the readability of the paper. Excellent suggestions.
>
> Indeed, the flow will be improved (and more consistent) if Section 2, Section 3.1, and Section 3.2 follow the same order.
>
> Our work is mainly motivated by the practical significance of minhash & its non-binary extension. Since MinHash & extensions are widely used in industry, especially in NLP, search and e-commence, we find developing DP for minhash and extensions will have a high impact.
>
> Let us describe one more application in LLM.  RAG can be done by either dense embeddings or sparse embeddings (ngrams or weighted n-grams) , or both. The embedding vectors can be shipped from the device to the cloud (or shared with other applications or different users in the same cloud system), but the privacy concerns must be addressed (because embeddings leak privacy). Therefore, our work provides a significant and practical solution. The embeddings can be generated on device (for example, for the coding agents) and the DP-protected vectors can be sent to the cloud, so that the coding agent can conduct RAG on the cloud.  We accidentally revealed a commercial application of the proposed method.
>
> From the academic perspective, we developed the new family of DP algorithms (even though some of them look quite simple), provided the theoretical analysis, and justified the algorithms by experiments.  We appreciate all the suggestions and will be happy to incorporate them in the revision or future research. Thanks again.

---

### Official Review · Reviewer_6Wxa · 2025-10-31

**Soundness:** 2
**Presentation:** 2
**Contribution:** 2
**Rating:** 4
**Confidence:** 4

**Summary:**

This paper extends OPH to suit the DP setting by introducing a random permutation strategy called DP-OPH. By theoretically validating the DP-OPH's DP ability, it can provide privacy protection of OPH and empirically present better performance compared with DP-MH.

**Strengths:**

1. Theoretical proof of the privacy protection property of DP-OPH.
2. Experiments showing the superior retrieval performance of DP-OPH over DP-MH.

**Weaknesses:**

1. The novelty is limited. This paper only extends the privacy ability of OPH. The proof seems to be a little bit similar to previous papers.
2. Lack of experiments on more bits.
3. The better retrieval performance seems to be benefited from the introduction of OPH not the novel privacy protection strategy as the retrieval performances of DP-OPH-fix and DP-MH on Webspam are really close.
4. The empirical justifications are limited. More experiments on CIFAR10, CIFAR100, MSCOCO, and Flickr30k would be great. Better empirically performance can only be validated by thoroughly experiments.
5. $\epsilon$ is too big for privacy concern. In practice, $\epsilon$ should be less than 0.1.
6. [1,2] are two practical post-hoc methods (Randomized Response) for DP-Hashing. Please compare with them for justify the performance and privacy protection.
7. The robust ϵ-DP analysis for the densified variants (DP-OPH-fix and DP-OPH-re) is heavily reliant on the assumption of a lower bound (f) on the number of non-zero elements in the data vector. If the true sparsity is heterogeneous or if f is set too high (excluding sparse data points), the DP guarantee either loosens or the utility suffers significantly by unnecessarily removing data points.


[1] Stanley L. Warner. Randomized response: A survey technique for eliminating evasive answer bias. Journal of the American Statistical Association, 60(309):63–69, 1965.

[2] Yimu Wang, Shiyin Lu, and Lijun Zhang. 2020. Searching Privately by Imperceptible Lying: A Novel Private Hashing Method with Differential Privacy. In Proceedings of the 28th ACM International Conference on Multimedia (MM '20).

**Questions:**

1. Please add the performance of OPH and MH (without random perturbation) to see why DP-OPH gets better performance. I do not get which partly contributes the most, the ‘new approach’ OPH or the random perturbation strategy.
2. Please add more experiments on bigger datasets and more bits to justify the superior performance of DP-OPH.
3. Please add experiments with adequate and practical choices of $\epsilon$
4. Please compare with [1,2] mentioned in Weakness.
5. Why choose the random permutation strategy? There are some other strategies [1,2 in Weakness] for the binary case. Please compare them theoretically and empirically.
6. See weakness

---

> ### Author Response · Authors · 2025-12-04
> **Reply to Reviewer 6Wxa**
>
> Thank you Reviewer 6Wxa:
>
> Let’s first clarify a major discrepancy about $\epsilon$. In your review
>
> *5. $\epsilon$ is too big for privacy concern. In practice,  $\epsilon$ should be less than 0.1.*
>
> Industry consensus seems to be that $\epsilon = 10$ is the sweet spot. The classical work https://arxiv.org/pdf/2303.00654  page 34, says
>
> - ε ≈ 10 seems to be a “sweet spot”
> - it gives acceptable utility for complex ML models
> - Empirical results show ε ≈ 10 is what works for DP-SGD
> - ε ≈ 10 corresponds to a very small probability of change in outcomes
> - This matches what Feldman et al. (2018) discuss
>
> Our paper is the first systematic work on the differential privacy of the family of MinHash-type algorithms. Given the widespread applications of minhash in industry, we believe the analysis on the new DP-OPH methods and the empirical justifications make novel and practical contributions to the community. Below we will answer the reviewer's specific questions.
>
> Q1. We have actually included the performance of non-private OPH and MH in the figures. The non-private OPH performance is plotted as the horizontal dotted lines in Figures 3, 4. For non-private MH, the performance is essentially the horizontal line matching the right endpoint of the DP-MH curves (when $\epsilon$ gets large, DP-MH converges to non-DP method). We did not plot it  to avoid overlapping lines. In most figures, non-private OPH and non-private MH performs very similarly. Therefore, the advantage of DP-OPH-re and DP-OPH-rand comes from our DP algorithm/analysis. In fact, we present Fig 1 to demonstrate improvement in "pure" privacy protection introduced by our proposed algorithms. Both DP-OPH-fix and DP-OPH-re have smaller privacy discount factor than DP-MH, which indicates that less noise is required for DP-OPH to achieve the same privacy as DP-MH.
>
> Q2&Q3. Our experiments cover $b=1,2,4$. These are common choices in most applications to reach the best efficiency-utility balance. Moreover, larger $b$ requires higher flipping probabilities thus more DP noise, so there is an "optimal" $b$ in the middle. In our experiments we found $b=2$ or $b=4$ usually provides best utility at a given privacy level. We are happy to add more discussion about this in the paper.
>
> We chose gene expression and web search datasets because these are two popular fields where minhash is heavily applied, as we cited in the paper, e.g.
>
> Ondov, et al. Mash: fast genome and metagenome distance estimation using minhash. Genome biology.
>
> Brown et al: a library for minhash sketching of DNA. J. Open Source Softw., 1(5):27, 2016
>
> and that minhash was originally designed for large-scale web search as in the seminal paper by Broder. (SEQUENCES), 1997.
>
> In fact, the popular webspam benchmark contains 350k samples which is larger than the image datasets suggested by the reviewer. We re-iterate that minhash-type methods are usually intended for high-dimensional sparse binary data.
>
> Moreover, the Appendix has the extension of the proposed DP-OPH methods to non-binary vectors called DP-BCWS. We experimented with the DailySports and MNIST datasets on training ML models. Our results show DP-BCWS is able to achieve very good accuracy with strong privacy protection. We hope this also interests the reviewer and partially addresses the question.
>
> Q4. Thanks for referring to the paper [1 ,2].   [1] from 1965 is the classical textbook work on randomized response. To serve the same purpose, we cited the review paper by Cynthia Dwork and Aaron Roth 2014.  [2] is an application of randomized response using a general formula for the flipping probability. Our work is different in that our analysis is specifically tied to OPH which is itself a randomized algorithm. The key design question is to determine the best flipping probability of the hash values, as we present in Lemma 3.1 and Theorem 3.2 to Theorem 3.4. This requires non-trivial statistical analysis and is the main theoretical contribution of our work. If the reviewer hoped us to expand the reference list on applications, we can cite more works.
>
> Q5. Our algorithms focus on the most fundamental (and challenging) privacy setting that protects the raw data vectors directly. As the hash values of minhash (and OPH) are discrete bits, it is natural to adopt the flipping-based randomized response techniques (with new analysis). It is also possible to directly add noise to the raw vector and then generate the hash values, but we experimented with that and found the performance is very poor. We are happy to include some examples if suggested. Thanks.
>
> We believe our replies covers most concerns in "Weakness" section. In addition to the detailed reply on W5 (should $\epsilon$ be 0.1 or 10?) at the beginning, as we cited in the paper, Zhao et al. also used $\epsilon=33.5$ in the private count sketch method.
>
> We hope our reply answers your questions adequately. Please let us know if we can help address more questions. Thank you

---

### Meta-Review · Area_Chair_vz9z · 2026-01-08

**Summary:**

This paper studies differentially private hashing/sketching for Jaccard similarity, focusing on MinHash and One Permutation Hashing (OPH) and their variants. It combine differential privacy (DP) with OPH, and propose DP-OPH framework with three variants to deal with empty bins in OPH. Across reviews, there is a strong positive score from Reviewer UvV5, while mUNu (2/10) assigns a very low score. Overall, the paper appears borderline-to-accept based on the strong supportive review and generally soundness/clarity feedback, but would benefit from revision based on suggestions from reviewers.

**Reviewer Concerns:**

Some concerns were addressed, such as:

- ε Choice and DP Validity: The rebuttal convincingly justified using ε≈10 as a standard utility–privacy tradeoff (consistent with DP-SGD practice), addressing concerns about ε being “too large.”

- The authors clarified that gains stem from the DP-OPH analysis (smaller privacy discount factor, less noise), not from non-private baselines, resolving confusion about where improvements come from.

- Algorithmic and Presentation Clarity: Ordering issues, heavy theorem presentation, and clarity of algorithm flow were acknowledged and accepted for revision, satisfying presentation-focused reviewers.

- Handling of Sparsity Lower Bound (f) in Practice: Concrete strategies were provided (enforcing f, per-vector f, sparsity binning), which largely addressed concerns about unrealistic assumptions in DP analysis.

- Estimator / Variance Discussion Gaps Identified and Accepted: Missing unbiased estimator and variance analysis were acknowledged with commitment to incorporate them, resolving the concern in principle.


However, some concerns are still outstanding, including:

- Perceived Lack of Novelty (Core Risk)
   Some reviewers (notably mUNu, possibly 6Wxa) still view the work as “straightforward DP via randomized response,” and the rebuttal did not introduce new technical evidence to decisively overturn this perception.

- Experimental Breadth and Isolation: Requests for broader datasets, more bit settings, and clearer equal-ε / equal-noise comparisons across DP-OPH and DP-MH remain only partially addressed.

- DP-OPH-fix vs DP-MH Closeness on Webspam: While conceptually explained, reviewers may still want cleaner empirical isolation to understand why methods converge on some datasets.

- Generality of f-Enforcement Assumption: Although plausible for text/n-gram data, enforcing or estimating f may be less practical in streaming or heterogeneous-sparsity settings; clearer practitioner guidance or adaptive designs are still missing.

- Presentation Complexity in Main Theorems: Some heavy theoretical results (e.g., Theorem 3.6) are still flagged as too complex for the main body and require restructuring to avoid hurting accessibility and reviewer confidence.

**Reviewer Scores:**

Reviewer 6Wxa (initial 4): If they accept ε≈10 as standard practice and recognize the DP-OPH advantage is not merely “OPH vs MH” but lower required noise via a better privacy discount factor, a move to 6 is plausible. If the novelty and “need more datasets/bits/baselines” stance dominates, they likely stay at 4.

Reviewer UvV5: Because suggested changes are editorial/structural, their score likely remains 8.

Reviewer mUNu (inital 2): The reviewer acknowledged soundness and comprehensive evaluation, so with discussion they might reconsider the numeric score to better match their own text. If the reviewer strongly equate “straightforward DP + standard analysis” with low novelty, they may only raise slightly or stay at 2.

Reviewer yXfd (inital 6): The rebuttal directly answered all pointed practicality questions and gave actionable practitioner guidance. If clarified well in the revision, this reviewer could raise the score. Given low confidence, they might keep 6 unless revisions improve clarity.

---

### Decision · Program_Chairs · 2026-01-26

Reject